# A novel central nervous system-penetrating protease inhibitor overcomes human immunodeficiency virus 1 resistance with unprecedented aM to pM potency

Manabu Aoki[1,2,3,4,5], Hironori Hayashi[6], Kalapala Venkateswara Rao[7,8], Debananda Das[1], Nobuyo Higashi-Kuwata[6], Haydar Bulut[1], Hiromi Aoki-Ogata[1,2,3,4], Yuki Takamatsu[1], Ravikiran S Yedidi[1], David A Davis[9], Shin-ichiro Hattori[6], Noriko Nishida[10], Kazuya Hasegawa[11], Nobutoki Takamune[12], Prasanth R Nyalapatla[7,8], Heather L Osswald[7,8], Hirofumi Jono[13], Hideyuki Saito[13], Robert Yarchoan[9], Shogo Misumi[14], Arun K Ghosh[7,8], Hiroaki Mitsuya[1,2,3,4,6]*

[1]Experimental Retrovirology Section, HIV and AIDS Malignancy Branch, National Cancer Institute, National Institutes of Health, Bethesda, United States; [2]Department of Hematology, Kumamoto University Graduate School of Medical Sciences, Kumamoto, Japan; [3]Department of Rheumatology, Kumamoto University Graduate School of Medical Sciences, Kumamoto, Japan; [4]Department of Infectious Diseases, Kumamoto University Graduate School of Medical Sciences, Kumamoto , Japan; [5]Department of Medical Technology, Kumamoto Health Science University, Kumamoto, Japan; [6]National Center for Global Health and Medicine Research Institute, Tokyo, Japan; [7]Department of Medicinal Chemistry and Molecular Pharmacology, Purdue University, West Lafayette, United States; [8]Department of Chemistry, Purdue University, West Lafayette, United States; [9]Retroviral Disease Section, HIV and AIDS Malignancy Branch, National Cancer Institute, National Institutes of Health, Bethesda, United States; [10]Bioanalysis Group, Drug Metabolism and Analysis Department, Nonclinical Research Center, Drug Development Service Segment, LSI Medience Corporation, Tokyo, Japan; [11]Protein Crystal Analysis Division, Japan Synchrotron Radiation Research Institute, Hyogo, Japan; [12]Innovative Collaboration Organization, Kumamoto University, Kumamoto, Japan; [13]Department of Pharmacy, Kumamoto University Hospital, Kumamoto, Japan; [14]Department of Environmental and Molecular Health Sciences, Faculty of Medical and Pharmaceutical Sciences, Kumamoto University, Kumamoto, Japan

*For correspondence: mitsuyah@nih.gov

Competing interests: The authors declare that no competing interests exist.

**Abstract** Antiretroviral therapy for HIV-1 infection/AIDS has significantly extended the life expectancy of HIV-1-infected individuals and reduced HIV-1 transmission at very high rates. However, certain individuals who initially achieve viral suppression to undetectable levels may eventually suffer treatment failure mainly due to adverse effects and the emergence of drug-resistant HIV-1 variants. Here, we report GRL-142, a novel HIV-1 protease inhibitor containing an unprecedented 6-5-5-ring-fused crown-like tetrahydropyranofuran, which has extremely potent activity against all HIV-1 strains examined with $IC_{50}$ values of attomolar-to-picomolar concentrations, virtually no effects on cellular growth, extremely high genetic barrier against the emergence of drug-resistant variants, and favorable intracellular and central nervous system

penetration. GRL-142 forms optimum polar, van der Waals, and halogen bond interactions with HIV-1 protease and strongly blocks protease dimerization, demonstrating that combined multiple optimizing elements significantly enhance molecular and atomic interactions with a target protein and generate unprecedentedly potent and practically favorable agents.

DOI: https://doi.org/10.7554/eLife.28020.001

## Introduction

Presently available combination antiretroviral therapy (cART) for human immunodeficiency virus type 1 (HIV-1) infection and acquired immunodeficiency syndrome (AIDS) potently suppresses the replication of HIV-1 and significantly extends the life expectancy of HIV-1-infected individuals (*Samji et al., 2013*; *Marcus et al., 2016*). Indeed, mortality rates for HIV-1-infected individuals who are treated before they have substantial immunologic damage have become close to that of general population and cART has been shown to reduce sexual transmission of HIV-1 by 93–96% (*Cohen et al., 2011*; *Cohen et al., 2016*). Yet, at the end of 2015, ~36.7 million people were living with HIV-1 infection, and 2.1 million people were newly infected and 1.7 million people lost their lives only in 2015 (http://www.unaids.org/sites/default/files/media_asset/global-AIDS-update-2016_en.pdf). However, the cure of HIV-1 infection or AIDS is still elusive (*Deeks et al., 2016*) and moreover, our ability to provide effective long-term cART remains a complex issue, since many of those who initially achieve favorable viral suppression to undetectable levels eventually suffer treatment failure. Nevertheless, in regard to the propensity of HIV-1 to develop resistance to antiretroviral agents, protease inhibitors (PIs) generally have high genetic barrier against resistance. In particular, the latest U. S. Food and Drug Administration (FDA)-approved PI, darunavir (DRV), the only PI recommended for first line therapy (*Panel on Antiretroviral Guidelines for Adults and Adolescents, 2016*), has a favorable genetic barrier apparently because of its dual mechanism of action: (i) protease enzymatic inhibition activity and (ii) protease dimerization inhibition activity (*Koh et al., 2007*; *Koh et al., 2011*; *Hayashi et al., 2014*), and currently represents the most widely used PI for treating HIV-1-infected individuals. Nevertheless, the emergence of DRV-resistant HIV-1 variants has been reported both in vitro and in vivo, and patients with such DRV-resistant HIV-1 variants may experience treatment failure (*Koh et al., 2010*; *Mitsuya et al., 2007*; *De Meyer et al., 2009*). Furthermore, 12–21% of individuals receiving DRV-containing regimens discontinue the treatment due to adverse events, mainly diarrhea and virological failure (*Clotet et al., 2007*; *Madruga et al., 2007*; *Ortiz et al., 2008*). An alternative PI, atazanavir (ATV), also frequently causes indirect hyperbilirubinemia (*Lennox et al., 2014*). Dolutegravir (DTG), an integrase strand transfer inhibitor (INSTI), is also widely used and recommended in the therapy of HIV-1 infection. However, serious adverse events such as gastrointestinal disorders have been seen in as high as 11% of individuals receiving DTG (*Clotet et al., 2014*). In addition, cART has less benefit in patients with progressive neuropsychological impairment such as HIV-associated neurocognitive disorders (HAND), which occur in 40–83% HIV-1-infected individuals (*Heaton et al., 2010*), and these patients are less likely to achieve long-term virologic suppression (*Tozzi et al., 2005*; *Saylor et al., 2016*). Thus, novel antiretrovirals with more potent activity, greater specificity to HIV-1 and tolerability, higher genetic barrier to the emergence of drug-resistant HIV-1 variants, and central nervous system (CNS)-penetration capability are required.

In the present study, we report a newly generated, two fluorine atoms-containing novel HIV-1 protease inhibitor, GRL-142, which has unprecedentedly potent activity against a wide range of HIV-1 variants with $IC_{50}$ values of attomolar to subnanomolar concentrations, extremely high genetic barrier against the emergence of resistance variants, and favorable CNS-penetration capability. We also demonstrate the structural basis of the GRL-142's unprecedentedly potent activity.

## Results

### Identification of GRL-142, an unprecedentedly potent HIV-1 protease inhibitor

Since our first report of darunavir (DRV) in 2003 (*Koh et al., 2003*), we continued optimization based on the structure of DRV, seeking novel protease inhibitors (PIs) that are more potent against a

variety of existing multi-PI-resistant HIV-1 variants with greater safety, do not permit or substantially delay the emergence of HIV-1 variants resistant to the very PIs, and favorably penetrate into the CNS, and identified GRL-142. GRL-142 contains newly generated pharmacophores such as an unprecedented 6-5-5 ring-fused crown-like tetrahydropyranofuran (Crn-THF) as the P2-ligand, P1-bis-fluoro-benzyl (bis-Fbz), and P2'-cyclopropyl-amino-benzothiazole (Cp-Abt)(Figure 1).

GRL-139, a prototype to GRL-142, structurally resembles DRV but contains the Crn-THF moiety instead of the bis-THF of DRV (Figure 1), and exerts comparable antiviral activity against wild-type HIV-1 (cHIV$_{NL4-3}^{WT}$) as compared to DRV (Table 1) (Ghosh et al., 2017). However, GRL-139 failed to block the replication of three highly DRV-resistant HIV-1 variants (HIV$_{DRV}^{R}$s) that were selected by propagating in the presence of increasing concentrations of DRV and are highly resistant to all presently clinically available PIs including DRV and nucleos(t)ide-reverse-transcriptase inhibitors (NRTIs) such as tenofovir (TDF) (Koh et al., 2010; Aoki et al., 2015) (Supplementary file 1). By contrast, GRL-036 also resembles DRV but has the Cp-Abt moiety and shows an improved anti-HIV-1 profile, more effectively blocking the replication of HIV$_{DRV}^{R}$s than DRV (Table 1) (Ghosh et al., 2017). GRL-121 contains both Crn-THF and Cp-Abt moieties and more effectively blocked the replication of cHIV$_{NL4-3}^{WT}$ by about 10-fold compared to DRV (Ghosh et al., 2017). GRL-121 more potently suppressed the replication of all three HIV$_{DRV}^{R}$s. Interestingly, the addition of two fluorine atoms to

**Figure 1.** Chemical structure of compounds. Structures of DRV, GRL-139, GRL-036, GRL-121, and GRL-142. P2-Crn-THF and P2'-Cp-Abt moieties are shown in red and blue, respectively.
DOI: https://doi.org/10.7554/eLife.28020.002

**Table 1.** Antiviral activity of novel four compounds against highly DRV-resistant HIV-1 variants.

| | Mean $IC_{50}$ in nM $\pm$ SD (fold-change) | | | | | | |
|---|---|---|---|---|---|---|---|
| | LPV | ATV | DRV | GRL-139 | GRL-036 | GRL-121 | GRL-142 |
| $cHIV_{NL4-3}^{WT}$ | 13 $\pm$ 2 | 4.0 $\pm$ 2.3 | 3.2 $\pm$ 0.7 | 2.8 $\pm$ 0.8 | 1.9 $\pm$ 0.2 | 0.26 $\pm$ 0.05 | 0.019 $\pm$ 0.017 |
| $HIV_{DRV}^{R}P20$ | >1000 (>77) | 450 $\pm$ 20 (113) | 51 $\pm$ 3 (16) | 36 $\pm$ 8 (13) | 5.9 $\pm$ 6 (3) | 0.075 $\pm$ 0.058 (0.3) | 0.0024 $\pm$ 0.002 (0.1) |
| $HIV_{DRV}^{R}P30$ | >1000 (>77) | >1000 (>250) | 220 $\pm$ 40 (79) | 350 $\pm$ 10 (125) | 28 $\pm$ 2 (15) | 1.9 $\pm$ 0.1 (7) | 0.023 $\pm$ 0.018 (1) |
| $HIV_{DRV}^{R}P51$ | >1000 (>77) | >1000 (>250) | 2500 $\pm$ 100 (781) | >1000 (>357) | 530 $\pm$ 70 (279) | 32 $\pm$ 4 (123) | 1.2 $\pm$ 2 (63) |

Numbers in parentheses represent fold changes in $IC_{50}$s for each isolate compared to the $IC_{50}$s for wild-type $cHIV_{NL4-3}^{WT}$. All assays were conducted in triplicate, and the data shown represent mean values ($\pm$1 standard deviation) derived from the results of three independent experiments.

DOI: https://doi.org/10.7554/eLife.28020.003

GRL-121, generating GRL-142 further strengthened the activity against $cHIV_{NL4-3}^{WT}$ achieving an $IC_{50}$ value as low as 0.019 nM compared to the 9 FDA-approved PIs of which $IC_{50}$s range from 3.2 to 330 nM (*Table 2*). GRL-142 had a much improved cytotoxicity profile with a selectivity index ($CC_{50}$/$IC_{50}$) as high as 2,473,684 (*Table 2*). GRL-142 also highly potently blocked the replication of all three $HIV_{DRV}^{R}$s by factors of 27–83 compared to GRL-121 (*Table 1*), while it should be noted that the most DRV-resistant HIV-1 variant, $HIV_{DRV}^{R}P51$, is less sensitive to GRL-142 by 63-fold than $cHIV_{NL4-3}^{WT}$, indicating considerable resistance shift. Notably, the $IC_{50}$ value of GRL-142 against $HIV_{DRV}^{R}P51$ (1.2 nM), the most multi-PI/NRTI-resistant $HIV_{DRV}^{R}$, was ~3 fold lower than that of DRV against $cHIV_{NL4-3}^{WT}$ (3.2 nM). We further examined the activity of GRL-142 against two HIV-2 strains and found that GRL-142 also exerts highly potent antiviral activity against the HIV-2 strains examined (*Table 2*).

We further examined the activity of five PIs including GRL-121 and -142 against seven resistant HIV-1 variants, which we had previously selected in vitro with each of the seven FDA-approved PIs ($_{invitro}HIV_{PI}^{R}$s: *Table 3*) (*Koh et al., 2010*; *Aoki et al., 2009*; *Aoki et al., 2012*). Most of the seven variants were significantly less susceptible to two PIs, lopinavir (LPV) and ATV (*Table 3*), that have presently been relatively well used in clinics. DRV also failed to effectively block most of the seven variants with $IC_{50}$ value fold-differences ranging from 2- to 86-fold. However, GRL-121 showed extremely potent activity against all the seven variants examined, presenting $IC_{50}$ values ranging 0.0018 to 0.13 nM. The activity of GRL-121 against all the seven variants was significantly more potent than that against $cHIV_{NL4-3}^{WT}$. Surprisingly, GRL-142 showed even more potent activity against the seven variants with $IC_{50}$ values of 0.0000019 nM (1.9 fM) to 0.015 nM. The activity of GRL-142 against the variants was also significantly more potent than that against $cHIV_{NL4-3}^{WT}$. We,

**Table 2.** Antiviral activity of GRL-121 and −142 against $cHIV_{NL4-3}^{WT}$ and two HIV-2 strains and their cytotoxicities in vitro.

| | Mean $IC_{50}$(nM) $\pm$ SD | | | | |
|---|---|---|---|---|---|
| Drug | $cHIV_{NL4-3}^{WT}$ | $HIV-2_{ROD}$ | $HIV-2_{EHO}$ | $CC_{50}$ ($\mu$M) | Selectivity index* |
| SQV | 12 $\pm$ 3 | 9.0 $\pm$ 5.0 | 8.8 $\pm$ 2.1 | 33 | 2750 |
| IDV | 18 $\pm$ 5 | 31 $\pm$ 3 | 55 $\pm$ 25 | 75 | 4167 |
| NFV | 23 $\pm$ 5 | 27 $\pm$ 0.4 | 84 $\pm$ 20 | 32 | 1391 |
| RTV | 34 $\pm$ 10 | 136 $\pm$ 165 | 278 $\pm$ 88 | 35 | 1.029 |
| TPV | 330 $\pm$ 13 | 293 $\pm$ 45 | 313 $\pm$ 48 | 34 | 103 |
| APV | 26 $\pm$ 8 | 170 $\pm$ 82 | 305 $\pm$ 78 | >150 | >4167 |
| LPV | 13 $\pm$ 8 | 40 $\pm$ 28 | 11 $\pm$ 2 | 33 | 2538 |
| ATV | 4.0 $\pm$ 2.3 | 28 $\pm$ 6 | 10 $\pm$ 8 | 80 | 20,000 |
| DRV | 3.2 $\pm$ 0.7 | 8.5 $\pm$ 0.7 | 6.2 $\pm$ 0.7 | 133 | 41,562 |
| GRL-121 | 0.26 $\pm$ 0.05 | 0.020 $\pm$ 0.014 | 0.071 $\pm$ 0.071 | 34 | 130,769 |
| GRL-142 | 0.019 $\pm$ 0.017 | 0.00032 $\pm$ 0.00015 | 0.000059 $\pm$ 0.000025 | 47 | 2,473,684 |

*Each selectivity index denotes a ratio of $CC_{50}$ to $IC_{50}$ against $cHIV_{NL4-3}^{WT}$.

The data shown represent mean values ($\pm$1 standard deviation) derived from the results of three independent experiments.

DOI: https://doi.org/10.7554/eLife.28020.004

**Table 3.** Antiviral activity of GRL-121 and −142 against highly PI-resistant HIV-1 variants.

| Virus species | | Mean IC$_{50}$ in nM ± SD (fold-change) | | | | |
| --- | --- | --- | --- | --- | --- | --- |
| | | LPV | ATV | DRV | GRL-121 | GRL-142 |
| Wild-type | cHIV$_{NL4-3}$$^{WT}$ | 13 ± 2 | 4.0 ± 2.3 | 3.2 ± 0.7 | 0.26 ± 0.05 | 0.019 ± 0.017 |
| $_{invitro}$HIV$_{PI}$$^{R}$* | HIV$_{SQV-5μM}$ | >1000 (>77) | 430 ± 20 (108) | 17 ± 7 (5) | 0.026 ± 0.01 (0.1) | 0.00018 ± 0.00003 (0.009) |
| | HIV$_{APV-5μM}$ | 280 ± 15 (22) | 3.0 ± 1.0 (1) | 39 ± 16 (12) | 0.13 ± 0.08 (0.5) | 0.0000085 ± 0.000008 (0.0004) |
| | HIV$_{LPV-5μM}$ | >1000 (>77) | 46 ± 10 (12) | 280 ± 50 (86) | 0.0018 ± 0.0006 (0.007) | 0.0000019 ± 0.0000014 (0.0001) |
| | HIV$_{IDV-5μM}$ | 250 ± 15 (19) | 56 ± 9 (14) | 37 ± 8 (12) | 0.0092 ± 0.0163 (0.04) | 0.00018 ± 0.00028 (0.009) |
| | HIV$_{NFV-5μM}$ | 37 ± 3 (3) | 12 ± 2 (3) | 7.7 ± 3 (2) | 0.048 ± 0.018 (0.2) | 0.00024 ± 0.00026 (0.01) |
| | HIV$_{ATV-5μM}$ | 310 ± 20 (24) | >1000 (>250) | 25 ± 1 (8) | 0.092 ± 0.097 (0.4) | 0.015 ± 0.004 (0.8) |
| | HIV$_{TPV-15μM}$ | >1000 (>77) | >1000 (>250) | 40 ± 3 (13) | 0.063 ± 0.016 (0.2) | 0.00024 ± 0.00007 (0.01) |
| $_{rCL}$HIV** | $_{rCL}$HIV$_{F16}$ | >1000 (>77) | 193 ± 23 (48) | 3357 ± 600 (1,049) | 2.5 ± 0.9 (10) | 0.016 ± 0.016 (0.8) |
| | $_{rCL}$HIV$_{F39}$ | >1000 (>77) | 374 ± 23 (94) | 313 ± 230 (98) | 0.028 ± 0.02 (0.1) | 0.0061 ± 0.002 (0.3) |
| | $_{rCL}$HIV$_{V42}$ | >1000 (>77) | 270 ± 20 (68) | 343 ± 28 (107) | 3.1 ± 1.9 (12) | 0.026 ± 0.023 (1) |
| | $_{rCL}$HIV$_{T44}$ | >1000 (>77) | >1000 (>250) | 2487 ± 210 (777) | 12 ± 6 (46) | 0.69 ± 0.66 (36) |
| | $_{rCL}$HIV$_{M45}$ | >1000 (>77) | >1000 (>250) | 1924 ± 1570 (601) | 3.8 ± 0.5 (15) | 0.094 ± 0.113 (5) |
| | $_{rCL}$HIV$_{T48}$ | >1000 (>77) | 440 ± 26 (110) | 315 ± 61 (98) | 1.1 ± 1.1 (4) | 0.0052 ± 0.0017 (0.3) |

*$_{invitro}$HIV$_{PI}$$^{R}$, in vitro PI-selected HIV-1 variants; **$_{rCL}$HIV, recombinant clinical HIV-1 variants.

Numbers in parentheses represent fold changes in IC$_{50}$s for each isolate compared to the IC$_{50}$s for wild-type cHIV$_{NL4-3}$$^{WT}$. All assays were conducted in triplicate, and the data shown represent mean values (±1 standard deviation) derived from the results of at least three independent experiments.

DOI: https://doi.org/10.7554/eLife.28020.005

furthermore, examined the activity of the five PIs against six recombinant infectious clinical HIV-1 variants ($_{rCL}$HIVs) that are highly resistant to all the currently available PIs including DRV (*Mitsuya et al., 2007*; *Aoki et al., 2015*). GRL-121 exerted highly potent activity with IC$_{50}$ values of 0.028 to 12 nM, while GRL-142 again showed even more potent activity with IC$_{50}$ values of 0.0052 to 0.69 nM (*Table 3*). Moreover, we carried out assays of DRV, GRL-121, and GRL-142 against 18 recombinant HIV-1 variants carrying a single amino acid substitution known to be associated with HIV-1 resistance to various PIs (*Table 4*). DRV was effective against all the recombinant clones with IC$_{50}$ values of 0.29 to 4.7 nM. GRL-121 was again found to be highly potent against all the variants examined with IC$_{50}$ values of 0.015 to 350 pM. GRL-142 was even more potent against the variants as compared to GRL-121. It is noted that GRL-142 was extremely potent against three recombinant HIV-1 variants (cHIV$_{NL4-3}$$^{V32I}$, cHIV$_{NL4-3}$$^{G48V}$, and cHIV$_{NL4-3}$$^{I50V}$) with IC$_{50}$ values of 12, 36, and 93 attomolar (aM), respectively (*Table 4* and *Supplementary file 2*).

## Multiple mechanisms of GRL-142's potent anti-HIV-1 activity

Dimerization of HIV-1 protease (PR) subunits is an essential process for PR's acquisition of proteolytic activity, which plays a critical role in the maturation of HIV-1. Thus, it is thought that the disruption of the dimerization process can inhibit HIV-1 maturation (*Wlodawer et al., 1989*; *Kohl et al., 1988*). In this regard, DRV has been shown to effectively disrupt the dimerization of PR monomer subunits as determined by the fluorescence resonance energy transfer (FRET)-based HIV-1 expression assay (*Koh et al., 2007*) (*Figure 2A*). We thus determined whether GRL-142 exerted PR dimerization inhibition activity. In the absence of compounds, the mean CFP$^{A/B}$ ratio obtained was 1.16, indicating that PR dimerization clearly occurred (*Figure 2B*). However, the ratio was decreased to 0.91 in the presence of 100 nM DRV, demonstrating that DRV blocked the dimerization of the PR subunit. Interestingly, GRL-142 blocked the dimerization at much lower concentrations (0.1 nM) than DRV. These data strongly suggest that as in the case of DRV, GRL-142 has bimodal HIV-1 inhibition mechanism, (i) inhibition of HIV-1 PR dimerization and (ii) inhibition of enzymatic activity of PR.

We have previously demonstrated that PR monomer subunits initially interact at the active site interface, generating unstably dimerized PR subunits, and subsequently the termini interface interactions occur, completing the dimerization process. DRV binds in the proximity of the active site interface of PR and blocks PR subunits dimerization (*Hayashi et al., 2014*). Therefore, we asked whether

**Table 4.** Antiviral activity of GRL-121 and −142 against HIV-1 variants carrying single amino acid substitution in PR region.

| Infectious clone | Amino acid substitution in PR | Mean IC$_{50}$ ± SD (nM) | | |
|---|---|---|---|---|
| | | DRV | GRL-121 | GRL-142 |
| cHIV$_{NL4-3}$$^{WT}$ | none | 3.2 ± 0.7 | $(2.6 \pm 0.5)\times10^{-1}$ | $(1.9 \pm 1.7)\times10^{-2}$ |
| cHIV$_{NL4-3}$$^{L10F}$ | L10F | 3.3 ± 1.0 | $(3.4 \pm 1)\times10^{-2}$ | $(2.9 \pm 1.2)\times10^{-3}$ |
| cHIV$_{NL4-3}$$^{L24I}$ | L24I | 3.1 ± 0.6 | $(1.2 \pm 1.1)\times10^{-4}$ | $(2.9 \pm 1.6)\times10^{-5}$ |
| cHIV$_{NL4-3}$$^{D30N}$ | D30N | 4.7 ± 1.0 | $(2.0 \pm 3.0)\times10^{-2}$ | $(4.7 \pm 2.0)\times10^{-3}$ |
| cHIV$_{NL4-3}$$^{V32I}$ | V32I | $(3.0 \pm 1.0)\times10^{-1}$ | $(8.0 \pm 10.5)\times10^{-5}$ | $(1.2 \pm 1.6)\times10^{-8}$ |
| cHIV$_{NL4-3}$$^{L33F}$ | L33F | 2.8 ± 1.1 | $(3.5 \pm 2.5)\times10^{-1}$ | $(1.8 \pm 1.0)\times10^{-2}$ |
| cHIV$_{NL4-3}$$^{M46I}$ | M46I | 3.3 ± 0.2 | $(2.2 \pm 0.8)\times10^{-2}$ | $(2.2 \pm 1.5)\times10^{-3}$ |
| cHIV$_{NL4-3}$$^{I47V}$ | I47V | 3.0 ± 0.6 | $(3.2 \pm 0.7)\times10^{-2}$ | $(1.2 \pm 0.9)\times10^{-3}$ |
| cHIV$_{NL4-3}$$^{G48V}$ | G48V | $(2.9 \pm 0.6)\times10^{-1}$ | $(6.0 \pm 1.1)\times10^{-5}$ | $(3.6 \pm 6.0)\times10^{-8}$ |
| cHIV$_{NL4-3}$$^{I50V}$ | I50V | 2.7 ± 1.0 | $(1.5 \pm 2.4)\times10^{-5}$ | $(9.3 \pm 15.1)\times10^{-8}$ |
| cHIV$_{NL4-3}$$^{I54M}$ | I54M | 3.2 ± 0.8 | $(3.1 \pm 3.4)\times10^{-3}$ | $(1.9 \pm 2.3)\times10^{-4}$ |
| cHIV$_{NL4-3}$$^{I54L}$ | I54L | 3.3 ± 0.2 | $(3.2 \pm 0.3)\times10^{-1}$ | $(3.1 \pm 2.6)\times10^{-3}$ |
| cHIV$_{NL4-3}$$^{I54V}$ | I54V | 3.0 ± 0.4 | $(3.2 \pm 1)\times10^{-4}$ | $(2.7 \pm 1.1)\times10^{-5}$ |
| cHIV$_{NL4-3}$$^{L63P}$ | L63P | 2.3 ± 0.7 | $(2.5 \pm 2.6)\times10^{-2}$ | $(5.9 \pm 5.8)\times10^{-3}$ |
| cHIV$_{NL4-3}$$^{V82A}$ | V82A | 2.9 ± 0.1 | $(5.0 \pm 2.0)\times10^{-3}$ | $(3.6 \pm 10)\times10^{-4}$ |
| cHIV$_{NL4-3}$$^{V82I}$ | V82I | 4.0 ± 1.1 | $(1.8 \pm 1.8)\times10^{-1}$ | $(2.0 \pm 0.5)\times10^{-2}$ |
| cHIV$_{NL4-3}$$^{V82T}$ | V82T | $(5.7 \pm 2.0)\times10^{-1}$ | $(1.5 \pm 1.0)\times10^{-5}$ | $(3.1 \pm 4.0)\times10^{-6}$ |
| cHIV$_{NL4-3}$$^{I84V}$ | I84V | 2.7 ± 1.1 | $(1.7 \pm 0.9)\times10^{-3}$ | $(3.9 \pm 1.6)\times10^{-5}$ |
| cHIV$_{NL4-3}$$^{L90M}$ | L90M | 4.2 ± 0.5 | $(3.2 \pm 2.7)\times10^{-2}$ | $(5.8 \pm 0.9)\times10^{-4}$ |

All assays were conducted in triplicate, and the data shown represent mean values (±1 standard deviation) derived from the results of at least three independent experiments.

DOI: https://doi.org/10.7554/eLife.28020.006

GRL-142 binds to monomer subunits using electrospray ionization mass spectrometry (ESI-MS). As shown in *Figure 2C*, the ESI-MS spectra of PR containing D25N substitution (PR$^{D25N}$), which was folded in the presence of drugs revealed five peaks of differently charged ions in the range of mass/charge ratio ($m/z$) of 1,500–2,900. Since +5 charged monomer ions and +10 charged dimer ions have the same $m/z$ ($m/z$ = 2164.75 for PR$^{D25N}$), the greatest peak detected at $m/z$ 2164.75 was determined to represent two forms, a PR monomer and PR dimer. Thus, the five peaks represent a monomer, two dimers, and two monomer+dimers (*Figure 2C*-top panel). When unfolded PR$^{D25N}$ was re-folded in the presence of DRV, six additional significant peaks appeared, three for monomer+DRV, and three for dimer+DRV (*Figure 2C*-middle panel). When the same PR$^{D25N}$ was re-folded in the presence of GRL-142, six significant peaks appeared, representing three for monomer+GRL-142, and three for dimer+GRL-142 (*Figure 2C*-bottom panel). Each of the six additional peaks seen with GRL-142 appeared greater than those seen with DRV. When compared with the heights of the dimer+monomer peak rendered 1.0, the average height of the three peaks of DRV-bound monomers and that of the three peaks of GRL-142-bound monomers were 0.046 and 0.312, respectively; and the average height of the three peaks with DRV-bound dimers and that of the three peaks with GRL-142-bound dimers were 0.060 and 0.188, respectively. These data suggest that GRL-142 more tightly bound to monomers by 6.78-fold and to dimers by 3.13-fold than DRV and at least in part explain the reason GRL-142 much more strongly blocked PR dimerization than DRV.

We also examined the thermal stability of PR$^{D25N}$ in the presence of saquinavir (SQV), DRV, or GRL-142, using differential scanning fluorimetry (DSF). As illustrated in *Figure 2D*, the T$_m$ value of PR$^{D25N}$ alone was 54.92°C, while in the presence of SQV and DRV, the values increased to 58.14°C and 58.21°C, respectively, suggesting that the thermal stability of PR$^{D25N}$ increased when SQV and DRV bound to PR$^{D25N}$. However, in the presence of GRL-142, the T$_m$ value of PR$^{D25N}$ turned out to be substantially high at 65.65°C and the difference in T$_m$ values between PR$^{D25N}$ alone and GRL-142-

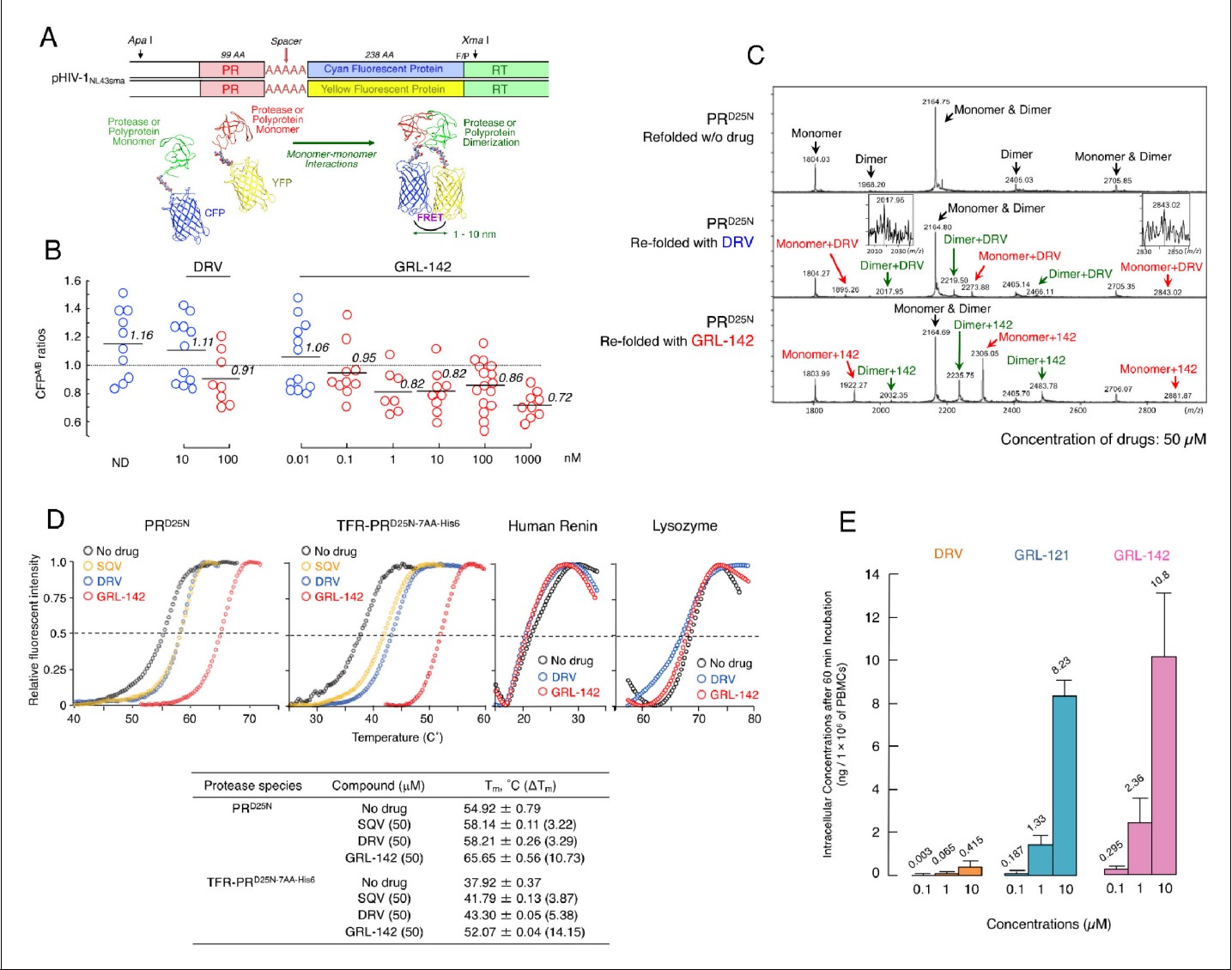

**Figure 2.** GRL-142 has significantly more potent HIV-1 protease dimerization inhibition activity, much greater thermal stability, and much higher intracellular concentration than DRV. (**A**) The FRET-based HIV-1 expression assay system detecting HIV-1 PR dimerization and its disruption by DRV or GRL-142. In an attempt to elucidate the dynamics of HIV-1 PR dimerization and the mechanisms of the emergence of HIV-1 resistance against certain PR inhibitors (PIs), we developed an intermolecular FRET-based HIV-1 expression assay system using CYP- and YFP-tagged PR monomers. Using this FRET-based HIV-1 expression assay system, we have previously identified nonpeptidyl small-molecule inhibitors of HIV-1 PR dimerization, including DRV (*Koh et al., 2007*; *Aoki et al., 2012*). Various plasmids encoding full-length molecular infectious HIV-1 (HIV-1$_{NL4-3}$) clones producing CFP- or YFP-tagged PR using the PCR-mediated recombination method were prepared. A linker consisting of five alanines was inserted between PR and fluorescent protein. A phenylalanine-proline site (F/P) that HIV-1 PR cleaves was also introduced between the fluorescent protein and reverse transcriptase. (Lower) Structural representations of PR monomers and dimer in association with the linker atoms and fluorescent proteins. FRET occurs only when the fluorescent proteins are 1–10 nm apart. (**B**) GRL-142 more potently inhibits PR$^{WT}$ dimerization by a factor of ~1000 than DRV. COS7 cells were exposed to various concentrations (0.01 to 1000 nM) of GRL-142 or DRV and were subsequently co-transfected with two plasmids, pHIV-PR$^{WT-CFP}$ and pHIV-PR$^{WT-YFP}$, respectively. After 72 hr, cultured cells were examined in the FRET-based HIV-1 expression assay and the CFP$^{A/B}$ ratios (Y axis) were determined. The arithmetic mean values of the ratios obtained are shown as horizontal bars. A CFP$^{A/B}$ ratio that is greater than one signifies that protease dimerization occurred, whereas a ratio that is less than one signifies that the disruption of protease dimerization occurred (*Koh et al., 2007*; *Aoki et al., 2012*). All the experiments were conducted in a blind fashion. The P values were determined using the Wilcoxon rank-sum test (JMP software, SAS, Cary, NC) and were 0.6721 for the CFP$^{A/B}$ ratio in the absence of drug (CFP$^{A/B}_{No-Drug}$) versus the CFP$^{A/B}$ ratio in the presence of 10 nM DRV (CFP$^{A/B}_{10-DRV}$), 0.0262 for CFP$^{A/B}_{No-Drug}$ versus CFP$^{A/B}_{100-DRV}$, 0.2483 for CFP$^{A/B}_{No-Drug}$ versus CFP$^{A/B}_{0.01- GRL-142}$, 0.0585 for CFP$^{A/B}_{No-Drug}$ versus CFP$^{A/B}_{0.1-GRL-142}$, 0.0145 for CFP$^{A/B}_{No-Drug}$ versus CFP$^{A/B}_{1-GRL-142}$, 0.0042 for CFP$^{A/B}_{No-Drug}$ versus CFP$^{A/B}_{10-GRL-142}$, 0.0056 for CFP$^{A/B}_{No-Drug}$ versus CFP$^{A/B}_{100-GRL-142}$, and 0.0019 for CFP$^{A/B}_{No-Drug}$ versus CFP$^{A/B}_{1000-GRL-142}$. (**C**) The ESI-MS spectrum of PR$^{D25N}$ in the absence or presence of 50 μM of DRV or GRL-142. Addition of DRV yielded two DRV-bound monomers and two DRV-bound dimers, while three GRL-142-bound monomers and three GRL-142-

*Figure 2 continued on next page*

*Figure 2 continued*

bound dimers were seen. Note that GRL-142 more tightly binds to monomers and dimers than DRV (by 6.78- and 3.13-fold; see the text), explaining at least in part the reason GRL-142 much more strongly blocks PR dimerization than DRV. (D) Thermal stability of PR$^{D25N}$ and TFR-PR$^{D25N-7AA-His6}$ in the absence or presence of SQV, DRV, or GRL-142 was determined using the differential scanning fluorimetry. Tm (50% melting temperature) values were determined as the temperature at which the relative fluorescent intensity became 50%. Note that the thermal stability curves with GRL-142 significantly shifted to the higher temperature (to the right) than those with no agent, SQV, or DRV. The Tm values with GRL-142 were much higher than those with no agent, SQV, or DRV. The thermal stability curves of human renin and lysozyme did not shift at all with GRL-142 as compared with those with no agent, indicating highly specific binding of GRL-142 to PR$^{D25N}$ and TFR-PR$^{D25N-7AA-His6}$. (E) GRL-142 achieves markedly higher intracellular concentration compared with DRV. PBMCs were incubated with 0.1, 1, and 10 μM of DRV, GRL-121, or GRL-142 for 60 min and vigorously washed with PBS and intra-cellular concentrations of each compound were determined using LC/MS. Arithmetic mean values shown are from the data derived from three independent experiments.

DOI: https://doi.org/10.7554/eLife.28020.007

bound PR$^{D25N}$ reached as high as 10.73°C, suggesting that GRL-142 more strongly binds to PR$^{D25N}$ than SQV or DRV. When we asked whether DRV and GRL-142 bound to TFR-PR$^{D25N-7AA-His6}$, a His-tagged transframe precursor form of PR$^{D25N}$ that contains seven N terminus amino acids of reverse transcriptase (7AA; PISPIET), both DRV and GRL-142 clearly bound to TFR-PR$^{D25N-7AA-His6}$, although the binding of GRL-142 (52.07°C) appeared to be significantly greater than that of DRV (43.30°C). Considering that GRL-142 much more strongly binds to PR monomer subunits than DRV, as in the case of DRV (*Hayashi et al., 2014*), GRL-142's monomer binding should be involved in the Gag-Pol auto-processing inhibition and even more effective than that of DRV.

It is known that fluorination increases lipophilicity because the carbon-fluorine bond is more hydrophobic than the carbon-hydrogen bond, often enhancing cell membrane penetration and oral bioavailability of the compounds containing carbon-fluorine bond (*Böhm et al., 2004*). Thus, we also determined intracellular concentrations of GRL-142 after incubating human peripheral blood mono-nuclear cells (PBMCs) in the presence of GRL-142 for 60 min. As shown in *Figure 2E*, intracellular concentrations of GRL-142 were significantly higher than those of DRV as examined under the same conditions. Achieving such high intracellular concentrations of GRL-142, together with the effective inhibition of PR dimerization, likely contributes to the unprecedentedly potent activity of GRL-142 against HIV-1.

## Structural analysis of GRL-142

We determined the structural interactions of GRL-142 with wild-type HIV-1 protease (PR$^{WT}$) using X-ray crystallography. GRL-142 binds in the active site of PR$^{WT}$ in two distinct conformations (related by 180° rotation) with relative occupancies of 0.53 and 0.47. The structural description is derived from the interactions of the major conformation of GRL-142 with PR$^{WT}$. GRL-142 has a *Crn*-THF as the P2-ligand moiety and a Cp-Abt as the P2' ligand moiety (*Figure 1*). Both groups make critical interactions with amino acids spanning distinct regions of PR$^{WT}$ active site. The *Crn*-THF moiety has two oxygen atoms and they both form hydrogen bonding interactions with the backbone amides of D29 and D30 (*Figure 3A*). The thiazole nitrogen makes a hydrogen bond interaction with the back-bone NH of D30'. The P2' amino group forms polar interactions with the sidechain carboxylate of D30'. The carbonyl and sulfonyl oxygens have polar interactions with the PR flap residues I50 and I50' through a bridging water molecule. The transition state mimic hydroxyl group forms polar inter-actions with the catalytic aspartates D25 and D25'. There is another hydrogen bond interaction from the amide nitrogen of the carbamate moiety to the backbone carbonyl oxygen of G27. Many of these polar interactions with HIV protease are also seen in DRV complexed with PR$^{WT}$ (*Figure 3A*).

Besides various critical polar interactions, GRL-142 has significant non-bonded van der Waals (vdW) interactions with PR$^{WT}$. The *Crn*-THF (*Figure 3B*-top right panel) has more favorable vdW con-tacts with PR$^{WT}$ residues compared to the *bis*-THF moiety of DRV (*Figure 3B*-top left panel). While the *Crn*-THF moiety of GRL-142 mediates better contacts with D29, D30, V32, I47, and L76 residues compared to the *bis*-THF moiety of DRV, the Cp-Abt moiety also forms better vdW contacts with the residues of D30' of PR$^{WT}$ (*Figure 3B*-bottom right panel). In particular, the cyclopropyl reaches out in the periphery of the active site gaining more contacts with D30' and K45' compared to the aminobenzene moiety of DRV (*Figure 3B*-bottom left panel).

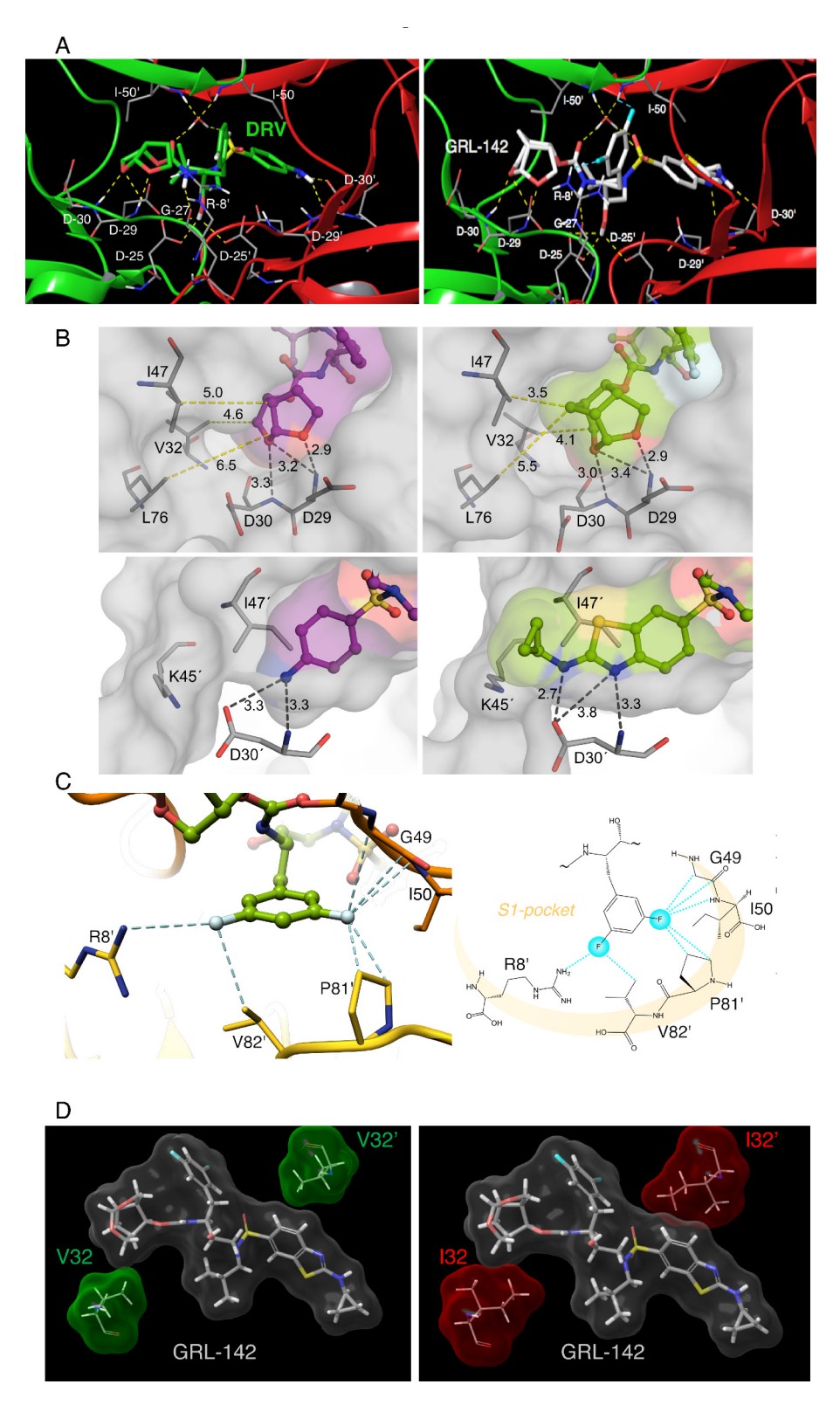

**Figure 3.** X-ray crystal structure analysis of GRL-142 with HIV-1 protease. (**A**) X-ray crystal structure of wild-type HIV-1 protease (PR^WT) in complex with DRV or GRL-142. The crystal structure of GRL-142 in complex PR^WT was solved (PDB ID: 5TYS). The polar interactions of GRL-142 with protease residues in the active site are shown, and the interactions of DRV with PR^WT (PDB ID: 4HLA) are shown for comparison. Cross-section of protease backbone is shown in green and red ribbons. The carbon atoms of GRL-142 and DRV are shown in off-white and green respectively. Nitrogen, oxygen, sulfur,

*Figure 3 continued on next page*

*Figure 3 continued*

fluorine, and hydrogen atoms are shown in blue, red, yellow, cyan, and white, respectively. Hydrogen bond interactions are shown by yellow dashed lines and polar interactions from fluorine are shown by cyan dashed lines. (B) Focus on the P2 and P2′ site interactions of DRV vs GRL-142. On the top panel, side by side comparison of *bis*-THF group of DRV (left panel) with *crn*-THF group of GRL-142 (right panel) in complex with HIV-1 protease. The *crn*-THF is larger with two two extra carbon atoms that contribute additional van der Walls interactions, particularly with three residues, I47, V32 and L76 in close proximity. On the lower panel, benzothioazole moiety of GRL-142 (right panel) with formation of extra ring effectively fills up the S2′subpocket. While sulfur atom forms close contact with I47′, cyclopropyl moiety protrudes outside the binding pocket and forms close contact with K45′. (C) Zoomed-in feature of the fluorine-mediated interactions inside the S1 pocket of dimerized PR$^{WT}$. The distances between the fluorine atoms and the interacting atoms less than 3.2 Å are shown as dashed cyan lines. While one fluorine of the *bis-meta*-fluorophenyl group is heavily involved in halogen bonding with G49, I50 and P81′, the other fluorine atom forms halogen bonds with the positively charged guanidinium group of R8′ as well as the side chain of V82′. The right panel scheme shows the halogen bonding within the S1 pocket as highlighted in light orange crescent shade. (D) GRL-142 has greater vdW contacts with I32 than with V32. The vdW interactions of GRL-142 with V32 and V32′ (green surfaces) in PR$^{WT}$ and with I32 and I32′ (red surfaces) in PR$^{V32I}$ mutant protease are shown. GRL-142 is shown in grey surface.

DOI: https://doi.org/10.7554/eLife.28020.008

The P2′ Cp-Abt of GRL-142 also has better contacts with the S2′ site of protease than DRV. We calculated the average vdW energy of interactions between GRL-142 or DRV with HIV-1 protease from molecular dynamics simulations (*Table 5*). The vdW energy of interactions between DRV and protease is −57.8 kcal/mol. GRL-142 has much better vdW interactions with protease as demonstrated by its lower interaction energy of −67.5 kcal/mol. The improved vdW interaction energy of ~10 kcal/mol may partly explain the much superior antiviral activity of GRL-142 compared to that of DRV. We also analyzed the vdW interaction energy with selected active site residues. GRL-142 has better interactions with Asp29/Asp29′ and Asp30/Asp30′ by ~3 kcal/mol. Interaction with these aspartates are important for the improved potency of PIs. Compared to DRV, GRL-142 also has a better interaction energy of ~3 kcal/mol with Ile47/Ile47′ which are located in the flap region of protease. We have observed that interactions with protease flap significantly improve the activity against not only wild-type protease but also against drug-resistant variants (*Aoki et al., 2015*).

The P1-phenyl moiety of GRL-142 has two fluorine atoms that form critical interactions with PR$^{WT}$ that are not formed by DRV. The crystal structure indicates that fluorine substitution causes very favorable halogen interactions within the S1-pocket of PR$^{WT}$ (*Figure 3C*). The high electronegativity of fluorine (3.98) *versus* hydrogen (2.20) leads to highly polarized halogen bond interactions with the backbone (F···H-C and F···C = O) of the G49 and I50, respectively. Contribution of fluorine atoms to the binding affinity of GRL-142 is not limited only by backbone interactions, but they also form close

**Table 5.** van der Waals interaction energies between GRL-142 or DRV with the protease dimer and selected active site residues.

|  | GRL-142 | DRV |
|---|---|---|
|  | (kcal/mol) | (kcal/mol) |
| Protease dimer | −67.5 | −57.8 |
| Asp29 & Asp29′ | −7.4 | −4.2 |
| Asp30 & Asp30′ | −5.4 | −2.9 |
| Val82 & Val82′ | −2.2 | −2.3 |
| Ile84 & Ile84′ | −4.3 | −4.6 |
| Ile47 & Ile47′ | −6.5 | −3.6 |
| Gly48 & Gly48′ | −3.3 | −2.8 |
| Gly49 & Gly49′ | −3.7 | −3.3 |
| Ile50 & Ile50′ | −8.7 | −9.2 |
| Pro81 & Pro81′ | −1.6 | −1.9 |
| Arg8 & Arg8′ | −2.2 | −2 |

Average van der Waals energies were calculated by analyzing the trajectories from a 1.2 ns molecular dynamics simulation using Desmond molecular dynamics system (D.E. Shaw Research, New York, NY 2017).

DOI: https://doi.org/10.7554/eLife.28020.009

contacts with the H-atoms of Cγ and Cδ of P81' (*Figure 3C*). Overall, the two fluorine atoms establish a halogen bond bridge within the S1-pocket by interacting with residues of both subunits of PR$^{WT}$. Moreover, direct interactions of GRL-142 with both R8' and I50 highly likely contribute significant additional polar interactions compared to other PIs. R8' and the PR's flap residues are associated with PR structure, functions, dynamics, substrate binding, and activity of PIs (*Koh et al., 2011*; *Yamazaki et al., 1994*).

We also compared the differences in vdW contacts of GRL-142 with V32I amino acid substitution, a key substitution for HIV-1's acquisition of PI-resistance (*Koh et al., 2010*; *Aoki et al., 2015*). As shown in *Figure 3D*, GRL-142 has good vdW contacts with wild-type V32; however, vdW interactions proved to be much better with mutated I32, especially the interactions of GRL-142 with I32'. Our simulations showed that I32 and I32' have an improved vdW interaction energy (~1 kcal/mol) with GRL-142 compared to V32 and V32'. These features should at least partly explain the significantly greater activity of GRL-142 to HIV-1 variants containing the V32I substitution in their PR.

## High genetic barrier to resistance of GRL-142

We also attempted to select HIV-1 variants resistant to ATV, DRV, GRL-121 and GRL-142 by propagating cHIV$_{NL4-3}$$^{WT}$ in MT-4 cells in the presence of increasing concentrations of each PI. When selected in the presence of ATV, the virus became resistant to the drugs and started replicating in the presence of 5 µM by 36 weeks (*Figure 4A*-top left panel). The PR-encoding region of the virus obtained from the culture concluded at week 36 contained seven amino acid substitutions, which were thought to be associated with the acquisition of HIV-1 resistance against ATV (*Figure 4—figure supplement 1A*). The development of DRV-resistant HIV-1 variants was significantly delayed and concentrations above 0.1 µM DRV still inhibited replication at week-36. However, cHIV$_{NL4-3}$$^{WT}$ failed to propagate in the presence of >0.01 µM of GRL-121 and >0.003 µM of GRL-142 even at week 36 (*Figure 4A*-top left panel), indicating that HIV-1 failed to select any meaningful resistant substitutions. The only substitutions that occurred were G16E and A71T (*Figure 4—figure supplement 1A*). Both these residues are located outside the active site of HIV-1 protease. When we employed a mixture of 11 multiple-PI-resistant clinical HIV-1 isolates (HIV$_{11MIX}$) as a starting HIV-1 population, the virus quickly became highly resistant to both ATV and DRV (*Figure 4A*-top right panel and *Figure 4—figure supplement 1B*), presumably due to the homologous recombination occurring within the HIV$_{11MIX}$ population (*Koh et al., 2010*). However, in the presence of >0.07 µM GRL-121 or >0.017 µM GRL-142, HIV$_{11MIX}$ failed to propagate. When we employed HIV$_{DRV}$$^R$$_{P30}$ and $_{rCL}$HIV$_{T48}$, both of which had been selected in the presence of DRV to be highly DRV-resistant (*Tables 1* and *3*), both virus populations again quickly became resistant to DRV (*Figure 4A*-bottom panels). In contrast, both populations failed to propagate in the presence of >0.01 µM GRL-142, although HIV$_{DRV}$$^R$$_{P30}$ became relatively resistant to GRL-121 but barely propagated in the presence of 0.3 µM GRL-121 at week 36 (*Figure 4A*-left bottom panel). This GRL-121-selected variant (HIV$_{DRV}$$^R$$_{P30}$$^{121-WK36}$) contained combination of three amino acid substitutions (L33F, I54M, and A82I) (*Figure 4—figure supplement 1C*), which are known to be associated with the loss of HIV-1 PR dimerization activity of DRV (*Koh et al., 2011*). HIV$_{DRV}$$^R$$_{P30}$$^{142-WK36}$ contained no significantly PI-resistance-associated amino acid substitutions; however, $_{rCL}$HIV$_{T48}$$^{142-WK36}$ contained various amino acid substitutions known to be associated with HIV-1 resistance against PIs including DRV (*Figure 4—figure supplement 1D*). Surprisingly, $_{rCL}$HIV$_{T48}$$^{142-WK36}$ contained even the five amino acid substitutions (V32I, L33F, I54L, V82A, and I84V) strongly associated with HIV-1's DRV resistance and the loss of DRV's PR dimerization inhibition activity (*Koh et al., 2011*), indicating that GRL-142 allows the persistence of various amino acid substitutions, yet does not permit HIV-1's acquisition of resistance to GRL-142. However, it is of note that while $_{rCL}$HIV$_{T48}$$^{142-WK36}$ had acquired resistance to GRL-142 by 769-fold compared to the starting population ($_{rCL}$HIV$_{T48}$)(*Table 6*), the absolute IC$_{50}$ value of GRL-142 against $_{rCL}$HIV$_{T48}$$^{142-WK36}$ was as low as 4.0 nM.

## CNS penetration of GRL-142 in rats

Persistent HIV-1 replication and inflammation in the CNS, which can occur even in patients receiving cART with an undetectable plasma viral load, is most likely responsible for HAND (*Saylor et al., 2016*). Hence, we finally quantified GRL-142 concentrations in plasma, cerebrospinal fluid (CSF), and brain of rats (n = 2) and compared those figures with those of DRV obtained under the same

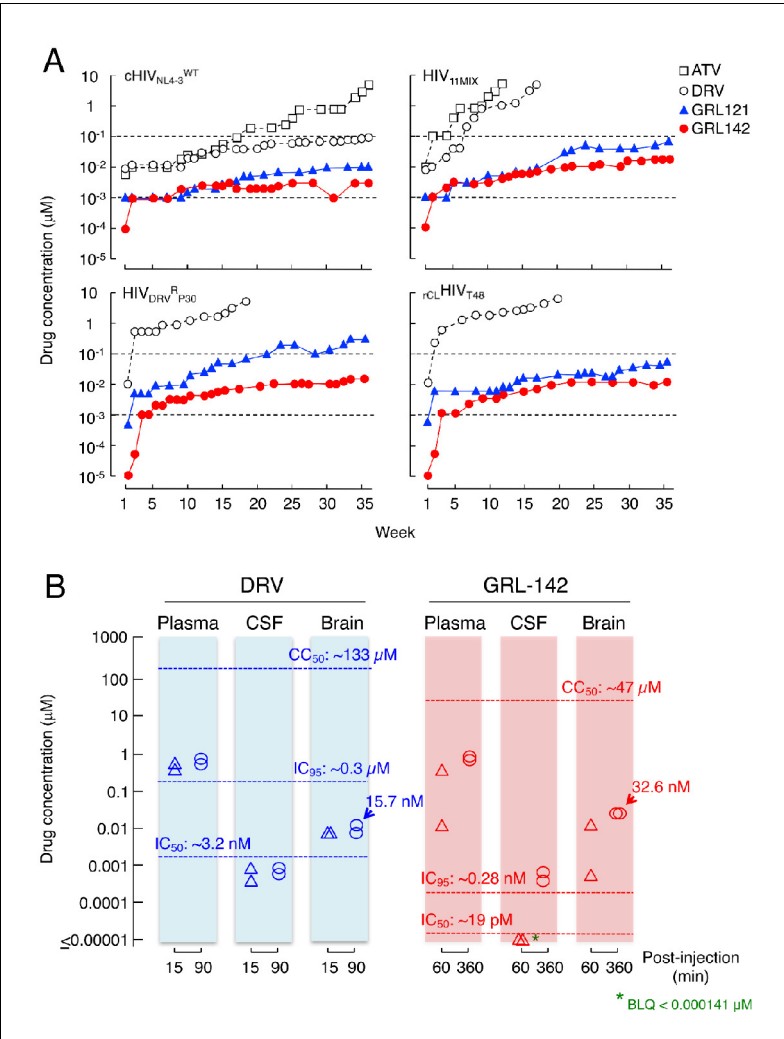

**Figure 4.** GRL-142 has an extremely high genetic barrier to the emergence of HIV-1 variants resistant to GRL-142 in vitro and effective penetration into brain in rats. (**A**) High genetic barrier of GRL-142 against the emergence of GRL-142-resistant variants. cHIV$_{NL4-3}$$^{WT}$ (top left panel), a mixture of 11 multi-PI-resistant HIV-1 isolates (HIV$_{11MIX}$) (top right panel), a DRV-resistant HIV-1 variant obtained from in vitro passage 30 with DRV (HIV$_{DRV}$$^{R}$$_{P30}$)(bottom left panel), and an infectious molecular HIV-1 clone derived from a heavily treated ART-experienced HIV-1-infected patient ($_{rCL}$HIV$_{T48}$)(bottom right panel) were propagated in the presence of increasing concentrations of each compound in MT-4 cells in a cell-free manner over 36 weeks. When HIV$_{11MIX}$, HIV$_{DRV}$$^{R}$$_{P30}$ and $_{rCL}$HIV$_{T48}$ were employed as a starting HIV-1 population, the virus quickly became highly resistant to ATV and DRV; however, all the starting virus populations failed to propagate in the presence of low concentrations of GRL-121 and GRL-142. GRL-142 did not allow the virus to acquire resistance and propagate more persistently than GRL-121. (**B**) Favorable penetration of GRL-142 into brain in rats. When DRV and GRL-142 were perorally administered to rats (n = 2) at a dose of 5 mg/kg plus RTV (8.33 mg/kg), the C$_{max}$ was achieved around 90 and 360 min after the administration, respectively. Thus, the concentrations of DRV and GRL-142 in plasma, CSF, and brain were determined in 15 and 90 min for DRV and 60 and 360 min for GRL-142 after the administration. The concentrations of GRL-142 in brain were 7.24 ± 9.65 and 32.6 ± 1.4 nM in 60 and 360 min, respectively. The latter concentration (32.6 nM) represents ~1,882 fold greater than the IC$_{50}$ value and ~114 fold greater than the IC$_{95}$ value of GRL-142. The concentrations of each compound were determined using LC-MS/MS and the lowest detection limit was 0.1 ng/ml (GRL-142: 14.1pM, DRV: 18.2 pM). These data strongly suggest that GRL-142 would potently block the infection and replication of HIV-1 in the brain.

DOI: https://doi.org/10.7554/eLife.28020.010

The following figure supplement is available for figure 4:

**Figure supplement 1.** Amino acid sequences of the protease-encoding regions of cHIV$_{NL4-3}$$^{WT}$, HIV$_{11MIX}$, HIV$_{DRV}$$^{R}$$_{P30}$, and $_{rCL}$HIV$_{T48}$in vitro selected with various agents.

*Figure 4 continued on next page*

*Figure 4 continued*

DOI: https://doi.org/10.7554/eLife.28020.011

conditions (*Figure 4B*). When DRV was perorally (PO) administered at a dose of 5 mg/kg together with RTV (8.33 mg/kg), the $C_{max}$ was achieved around 90 min after the PO administration. The DRV concentrations determined in 15 and 90 min after the peroral administration turned out to be 0.595 μM and 0.847 μM in plasma; 0.00100 μM and 0.00116 μM in CSF; and 0.0110 μM and 0.0157 μM in brain, respectively. The plasma concentrations of DRV were much greater than its $IC_{50}$ value (~3.2 nM); however, concentrations in CSF were lower than the $IC_{50}$ value and those in brain were slightly above the DRV $IC_{50}$ value but still substantially lower than the $IC_{95}$ value of DRV (~0.3 μM), suggesting that DRV likely fail blocking the replication of HIV-1 in the CNS. By contrast, when GRL-142 was perorally administered at a dose of 5 mg/kg together with RTV (8.33 mg/kg), the $C_{max}$ was achieved around 360 min after the PO administration. Plasma samples collected 60 and 360 min after the PO administration contained 0.189 μM and 0.974 μM GRL-142, respectively. Although CSF samples contained below detection levels (<0.000141 μM), brain contained 0.00724 μM and 0.0326 μM in 60 and 360 min, respectively. Since the $IC_{50}$ and $IC_{95}$ values of GRL-142 were ~19 pM and 0.28 nM, GRL-142 concentrations in brain are calculated to be ~1,882 fold greater than the $IC_{50}$ value of GRL-142 and ~114 fold greater than $IC_{95}$ of GRL-142, while ~562 fold lower than the $CC_{50}$ value of GRL-142. These data strongly suggest that GRL-142 would potently block the infection and replication of HIV-1 in brain.

## Discussion

GRL-142 should potentially serve as a most powerful agent in constructing future cART and overcome the issue of HIV-1's drug-resistance. Moreover, since highly potent new drugs such as the INSTIs and potent PIs (e.g., DRV) became available, the design of suppressive 'salvage' regimens has been possible for treating heavily antiretroviral therapy-experienced patients harboring multi-drug-resistant HIV-1 variants. However, such 'salvage' regimens yet remain challenging with high pill burden, high cost, more adverse effects, and greater risk of nonadherence, leading to virologic failure and viral acquisition of resistance. Thus, simplified antiretroviral regimens such as once-daily 2-tablet combination have been proposed. Indeed, one of such simplified regimens (elvitegravir/cobicistat/emtricitabine/tenofovir alafenamide plus DRV) has resulted in durable maintenance of virologic suppression, greater adherence and improved quality of life (*Huhn et al., 2017*). We posit that GRL-142, with its extremely high genetic barrier against the emergence of resistant HIV-1 variants (*Figure 4A*), would serve as a most favorable member of such simplified regimens. GRL-142 might also be used even for monotherapy, a possibly acceptable alternative for long-term clinical management of HIV-1 infection, although cautious and deliberate viral load monitoring and prompt reintroduction of cART as needed should be carried out.

The unprecedentedly potent antiviral activity of GRL-142 (picomolar to attomolar ranges) against a variety of HIV-1s including highly PI- and NRTI-resistant variants apparently results from markedly increased polar and non-polar interactions of the two novel moieties (*Crn*-THF and Cp-Abt) with critical amino acids of PR (*Figure 3A–C*), extensive halogen bonding networks established by two fluorines (*Figure 3C*), and highly potent inhibition of PR dimerization (*Figure 2B,C*). It also appears that the presence of two fluorines results in the increased cell membrane permeability (*Figure 2E*), which

**Table 6.** Antiviral activity of GRL-142 against viruses obtained after in vitro selection.

| Virus before selection | Mean $IC_{50}$ in nM $\pm$ SD | Virus after selection | Mean $IC_{50}$ in nM $\pm$ SD | Fold-change |
|---|---|---|---|---|
| cHIV$_{NL4-3}$$^{WT}$ | 0.019 ± 0.017 | HIV$_{NL4-3}$$^{142-WK36}$* | 0.29 ± 0.06 | 15 |
| rCL HIV$_{T48}$ | 0.0052 ± 0.0017 | rCL HIV$_{T48}$$^{142-WK36}$* | 4.0 ± 0.5 | 769 |

*Viruses were obtained at 36 week of the in vitro GRL-142 selection.

All assays were conducted in triplicate, and the data shown represent mean values (±1 standard deviation) derived from the results of three independent experiments.

DOI: https://doi.org/10.7554/eLife.28020.012

probably contributes to the potent activity of GRL-142. The V32I substitution closely associated with HIV-1 resistance to a variety of PIs further increased vdW contacts with $PR^{32I}$ than with $PR^{V32}$ (*Figure 3D*), strongly suggesting that GRL-142 effectively fills the active site cavity of mutated PR and exerts potent activity against various PI-resistant HIV-1 variants. This feature is likely associated with the observation that GRL-142 hardly allows HIV-1 to acquire resistance to GRL-142 under selection (*Figure 4A*). Furthermore, with the favorable CNS penetration capability combined with its potent activity, GRL-142 achieved ~1882 fold greater concentration than its $IC_{50}$ and ~114 fold greater concentration than its $IC_{95}$ in rat brain, strongly suggesting that GRL-142 would potently block the infectivity and replication of HIV-1 in brain (*Figure 4B*), although the efficacy of GRL-142 on the HIV-1-associated CNS abnormalities has to be examined in controlled clinical studies. Most importantly, the present data, taken together, indicate that combined multiple optimizing elements such as the unprecedented *Crn*-THF and Cp-Abt significantly enhance molecular and atomic interactions with a target protein and generate unprecedentedly potent and practically favorable agents.

## Materials and methods

### Antiviral agents

The synthetic methods of nonpeptidic PIs, GRL-036, -121, and -139 were recently published (*Ghosh et al., 2017*), while GRL-142 was newly designed and synthesized. The method of synthesis of GRL-142 will be published elsewhere by A. K. Ghosh *et al*. DRV was synthesized as previously described (*Ghosh et al., 2004*). Saquinavir (SQV), indinavir (IDV), nelfinavir (NFV), ritonavir (RTV), tipranavir (TPV), amprenavir (APV), lopinavir (LPV), atazanavir (ATV), zidovudine (AZT), lamivudine (3TC), and abacavir (ABC) were purchased from Sigma-Aldrich (St. Louis, MO). Tenofovir disoproxil fumarate (TDF) was purchased from BioVision (Milpitas, CA).

### Cells and viruses

MT-4 cells (RRID: CVCL_2632), authenticated using the STR profiling (ATCC, Manassas, VA), were grown in RPMI 1640-based culture medium, while COS7 cells (RRID: CVCL_0224) were propagated in Dulbecco's modified Eagle's medium. These media were supplemented with 10% fetal calf serum (FCS; PAA Laboratories GmbH, Linz, Austria) plus 50 U of penicillin and 50 μg of kanamycin per ml. Both cell lines were tested for mycoplasma contamination using the Universal Mycoplasma Detection Kit (ATCC). Peripheral blood mononuclear cells (PBMCs) were isolated from buffy coats obtained from HIV-1-seronegative individuals with Ficoll-Hypaque density gradient centrifugation and cultured at a concentration of $10^6$ cells/ml in RPMI 1640-based culture medium.

The following HIV-1 strains were used for the drug susceptibility assay and in vitro selection experiments: eleven HIV-1 clinical strains ($HIV_A$, $HIV_B$, $HIV_C$, $HIV_G$, $HIV_{TM}$, $HIV_{MM}$, $HIV_{JSL}$, $HIV_{SS}$, $HIV_{ES}$, $HIV_{EV}$, and $HIV_{13-52}$), which were originally isolated from patients with AIDS as previously described (*Tamiya et al., 2004*; *Yoshimura et al., 1999*; *Harada et al., 2007*). The clinical strains that contained 8 to 16 amino acid substitutions in the protease-encoding region (*Supplementary file 3*), which are associated with HIV-1 resistance to various PIs and have been genotypically and phenotypically characterized to be multi-protease inhibitor (PI)-resistant HIV-1 (25). Six recombinant clinical HIV-1 isolates ($_{rCL}HIVs$)($_{rCL}HIV_{F16}$, $_{rCL}HIV_{F39}$, $_{rCL}HIV_{V42}$, $_{rCL}HIV_{V44}$, $_{rCL}HIV_{T45}$, and $_{rCL}HIV_{T48}$) (*Mitsuya et al., 2007*; *Aoki et al., 2015*) were kindly provided by Robert Shafer of Stanford University, which were obtained using each of recombinant infectious HIV-1-encoding plasmids generated by ligating patient-derived amplicons encompassing approximately 200 nucleotides of Gag (beginning at the unique *Apa*I restriction site), the entire protease, and the first 72 nucleotides of reverse transcriptase-encoding gene using the expression vector pNLPFB (a generous gift from Tomozumi Imamichi of the National Institute of Allergy and Infectious Diseases). These recombinant clinical isolates contained 16 to 23 PI-resistance-associated amino acid substitutions in their protease-encoding region (*Supplementary file 2*). Eight HIV-1 variants resistant to PI that had been selected in vitro ($_{invitro}HIV_{PI}^R$s) with each of eight FDA-approved PIs (SQV, APV, LPV, IDV, NFV, ATV, TPV, and DRV) were also employed (*Koh et al., 2010*; *Aoki et al., 2009*; *Aoki et al., 2012*). All the variants used in the present study were confirmed to have acquired multiple amino acid substitutions in protease (*Supplementary file 3*), which have reportedly been associated with viral resistance to PIs. Each of the PI-selected variants ($HIV_{SQV-5μM}$, $HIV_{APV-5μM}$, $HIV_{LPV-5μM}$, $HIV_{IDV-5μM}$, $HIV_{NFV-5μM}$, $HIV_{ATV-5μM}$,

HIV$_{TPV-15\mu M}$, HIV$_{DRV}{}^R{}_{P20}$, HIV$_{DRV}{}^R{}_{P30}$, and HIV$_{DRV}{}^R{}_{P51}$) was highly resistant to the corresponding PI, with which the variant was selected against, and the differences in the IC$_{50}$s relative to the IC$_{50}$ of each drug against wild-type HIV-1 (cHIV$_{NL4-3}{}^{WT}$) ranged from 16- to 781-fold (*Table 3* and *Supplementary file 1*). HIV-1 clones carrying single amino acid substitutions were generated by site-directed mutagenesis using the QuikChange site-directed mutagenesis kit (Stratagene, La Jolla, CA) as previously described (*Gatanaga et al., 2002*). Determination of the nucleotide sequences of all plasmids constructed confirmed that each clone had the desired mutations but no unintended mutations. 293 T cells were transfected with each recombinant plasmid by using Lipofectamine 2000 reagent (Invitrogen, Carlsbad, CA). All the HIV-1 strains obtained were stored at −80°C until use.

## Antiviral and cytotoxicity assays

Antiviral assays were conducted as previously described (*Yoshimura et al., 1999*). Briefly, the designated concentrations of each compound tested were prepared by ten-fold serial dilution using the working solution of the compound (2 μM) directly in 96-well microtiter culture plates. MT-4 cells (10$^5$/ml) were exposed to 50% tissue culture infective dose (TCID$_{50}$) of each HIV-1 strain in the presence or absence of various concentrations of compound and incubated at 37°C. On day 7 of culture, the supernatant was harvested and the amount of p24 Gag protein was determined using the fully automated chemiluminescent enzyme immunoassay system (Lumipulse G1200; Fujirebio Inc., Tokyo, Japan). The drug concentrations that suppressed the production of p24 Gag protein by 50% (50% inhibitory concentrations; IC$_{50}$) were determined by comparison with the level of p24 production in drug-free control cell cultures. All assays were performed in triplicate. To determine the drug susceptibility of HIV-2$_{EHO}$ and HIV-2$_{ROD}$, MT-4 cells (10$^5$/ml) were exposed to 100 TCID$_{50}$ of HIV-2$_{EHO}$ or HIV-2$_{ROD}$ in the presence or absence of various concentrations of each compound, followed by incubation at 37°C for 7 days. The number of viable cells was determined using the Cell Counting Kit-8 (Dojindo, Kumamoto, Japan) and the magnitude of viral inhibition by each compound was determined based on their inhibitory effects of virally-induced cytopathicity in MT-4 cells.

For cytotoxicity, cells were plated in 96-well microtiter culture plates at a density of 10$^5$/ml and continuously exposed to various concentrations of each compound throughout the entire period of the culture. The number of viable cells in each well was determined using Cell Counting Kit-8. The 50% cytotoxic concentrations (CC$_{50}$) were determined as the concentration required to reduce the number of the cells by 50% compared to that of drug-unexposed control cultures.

## In vitro generation of highly GRL-142-resistant HIV-1 variants

We attempted to select HIV-1 variants resistant against GRL-121 and −142 as previously described (*Koh et al., 2010*; *Aoki et al., 2009*; *Aoki et al., 2012*). Briefly, thirty TCID$_{50}$ of each of eleven highly multi-PI-resistant HIV-1 isolates (HIV$_A$, HIV$_B$, HIV$_C$, HIV$_G$, HIV$_{TM}$, HIV$_{MM}$, HIV$_{JSL}$, HIV$_{SS}$, HIV$_{ES}$, HIV$_{EV}$, and HIV$_{13-52}$) was mixed and propagated in a mixture of an equal number of phytohemagglutinin (PHA)-stimulated peripheral blood mononuclear cells (PBMCs)(5 × 10$^5$) and MT-4 cells (5 × 10$^5$), in an attempt to adapt the mixed viral population for their replication in MT-4 cells. The cell-free supernatant harvested on day 7 of the co-culture was referred as HIV-1$_{11MIX}$. In the first passage, MT-4 cells (5 × 10$^5$) were exposed to the HIV-1$_{11MIX}$-containing supernatant or 500 TCID$_{50}$ of cHIV$_{NL4-3}{}^{WT}$, $_{rCL}$HIV$_{T48}$, or HIV$_{DRV}{}^R{}_{P30}$ and cultured in the presence of each compound at an initial concentration of a IC$_{50}$ dose. On the last day of each passage (week 1 to 3), 1.5 ml of the cell-free supernatant was harvested and transferred to a culture of fresh uninfected MT-4 cells in the presence of increased concentrations of the compound for the following round of culture. In this following round of culture, three drug concentrations (increased by one-, two-, and three-fold compared to the previous concentration) were employed. When the culture supernatant contained >200 ng/ml of p24 Gag protein, the HIV-1 isolate was assumed to have substantially replicated and the highest drug concentration among the three concentrations was used to continue the selection (for the next round of culture). This protocol was repetitively used until the drug concentration reached the targeted concentration (regularly 5 μM). Proviral DNA preparations obtained from the lysates of infected cells at indicated passages were subjected to nucleotide sequencing.

## Determination of nucleotide sequences

Molecular cloning and determination of the nucleotide sequences of HIV-1 strains passaged in the presence of each compound were performed as previously described (*Aoki et al., 2009*). In brief, high-molecular-weight DNA was extracted from HIV-1-infected MT-4 cells by using the InstaGene Matrix (Bio-Rad Laboratories, Hercules, CA) and was subjected to molecular cloning, followed by nucleotide sequence determination. The PCR primers used for the protease-encoding region were KAPA-1 (5'-GCAGGGCCCCTAGGAAAAAGGGCTGTTGG-3') and Ksma2.1 (5'-CCATCCCGGGC TTTAATTTTACTGGTAC-3'). The PCR mixture consisted of 1 µl proviral DNA solution, 10 µl Premix *Taq* (Ex *Taq* Version; Takara Bio Inc., Otsu, Japan), and 10 pmol of each PCR primer in a total volume of 20 µl. The PCR conditions used were an initial 1 min at 95°C, followed by 30 cycles of 30 s at 95°C, 20 s at 55°C, and 1 min at 72°C, with a final 7 min of extension at 72°C. The PCR products were purified using spin columns (illustra MicroSpin S-400 HR columns; GE Healthcare Life Science., Pittsburgh, PA), cloned directly, and subjected to sequencing using a model 3130 automated DNA sequencer (Applied Biosystems, Foster City, CA).

## Generation of FRET-based HIV-1 expression system

The intermolecular fluorescence resonance energy transfer-based HIV-1-expression assay employing cyan and yellow fluorescent protein (CFP and YFP, respectively)-tagged protease monomers was generated as previously described (*Koh et al., 2007*). In brief, CFP- and YFP-tagged HIV-1 protease-encoding plasmids were generated using BD Creator™ DNA Cloning Kits (BD Biosciences, San Jose, CA). For the generation of full-length molecular infectious clones containing CFP- or YFP-tagged protease, the PCR-mediated recombination method was used (*Fang et al., 1999*). A linker consisting of five alanines was inserted between protease and fluorescent proteins. The phenylala-nine-proline site where HIV-1 protease cleaves was also introduced between the fluorescent protein and reverse transcriptase. Thus obtained DNA fragments were subsequently joined by using the PCR-mediated recombination reaction performed under the standard condition for Ex*Taq* polymerase (Takara Bio Inc., Otsu, Japan). The amplified PCR products were cloned into pCR-XL-TOPO vector according to the manufacturer's instructions (Gateway Cloning System; Invitrogen, Carlsbad, CA). PCR products containing CFP- and YFP-encoding fragments were generated using pCR-XL-TOPO vector as templates, followed by digestion using both *Apa*I and *Xma*I, and the *Apa*I-*Xma*I fragment was introduced into pHIV-1$_{NLSma}$, generating pHIV-PR$^{WT-CFP}$ and pHIV-PR$^{WT-YFP}$, respectively.

## FRET procedure

COS7 cells plated on EZ view cover-glass bottom culture plates (Iwaki, Tokyo) were transfected with pHIV-PR$^{WT-CFP}$ and pHIV-PR$^{WT-YFP}$ using Lipofectamine 2000 (Invitrogen, Carlsbad, CA) according to manufacturer's instructions in the presence of various concentrations of each compound, cultured for 72 hr, and analyzed under Fluoview FV500 confocal laser scanning microscope (OLYMPUS Optical Corp, Tokyo) at room temperature as previously described (*Koh et al., 2007*). To determine whether each compound exerted HIV-1 protease dimerization, test compounds were added to the culture medium simultaneously with plasmid transfection. Results of FRET were determined by quenching of CFP (donor) fluorescence and an increase in YFP (acceptor) fluorescence (sensitized emission), since a part of the energy of CFP is transferred to YFP instead of being emitted. The changes in the CFP and YFP fluorescence intensities in the images of selected regions were quantified using Olympus FV500 Image software system (OLYMPUS Optical Corp). Background values were obtained from the regions where no cells were present and were subtracted from the values for the cells examined in all calculations. Ratios of intensities of CFP fluorescence after photobleaching to CFP fluorescence prior to photobleaching (CFP$^{A/B}$ ratios) were determined. It is well established that the CFP$^{A/B}$ ratios greater than 1.0 indicate that close contact (1–10 nm) of CFP- and YFP-tagged proteins occurred, indicating that the dimerization of protease subunits took place. When the CFP$^{A/B}$ ratios were less than 1, it indicates that the close contact of the two subunits did not occur, indicating that protease dimerization was inhibited (*Koh et al., 2007*).

## Expression and purification of protease species

Expression and purification of protease were carried out as previously described (*Hayashi et al., 2014*). Briefly, Rosetta (DE3) pLysS strain (Novagen) was transformed with an expression vector (pET-30a), which contained the genes of wild-type HIV-1$^{NL4-3}$-PR (PR$^{WT}$), or HIV-1-PR containing D25N substitution (PR$^{D25N}$) or His-tagged transframe precursor form of PR$^{D25N}$ carrying seven N terminus amino acids of reverse transcriptase (7AA; PISPIET)(TFR-PR$^{D25N-7AA-His6}$) using heat-shock. The culture was grown in a shake flask containing 30 mL of Luria broth plus kanamycin and chloramphenicol (LB$^{Km+/Cp+}$) at 37°C overnight. In the expression of PR$^{D25N}$ and TFR-PR$^{D25N-7AA-His6}$, twenty milliliters of the culture above was added to 1 L of LB$^{Km+/Cp+}$ and the LB$^{Km+/Cp+}$ culture was further grown to an optical density of 0.5 at 600 nm, and the expression was induced by adding 1 mM isopropyl β-D-thiogalactopyranoside for 3 hr. In the expression of PR$^{WT}$, twenty milliliters of the grown culture were added to 1 L of ZYM-10052 [1.0% N-Z amine, 0.5% yeast extract, 25 mM disodium hydrogen phosphate, 25 mM potassium dihydrogen phosphate, 50 mM ammonium chloride, 5 mM sodium sulfate, 1.0% glycerol, 0.05% glucose, 0.2% α-lactose, 2 mM magnesium sulphate] plus kanamycin and chloramphenicol (ZYM-10052$^{Km+/Cp+}$). The ZYM-10052$^{Km+/Cp+}$ culture was further continued at 37°C for 20 ~ 22 hr (*Studier, 2005*). Then the culture was spun down for pellet collection, and thus-obtained pellets were stored at −80°C until use. For purification of PRs, each pellet was resuspended in buffer A [20 mM Tris, 1 mM EDTA, and 1 mM DTT] and lysed with sonication. The cell lysates were separated into a supernatant fraction and an inclusion body fraction with centrifugation. The PRs were confirmed to be present in the inclusion body fraction, which was washed five times with buffer A containing 2 M urea and then with buffer A without urea. The twice-washed pellet was solubilized and PRs were unfolded with 100 mM formic acid (pH 2.8) (*Louis et al., 1999*). The unfolded PRs were purified using the fast protein liquid chromatography system (ÄKTA pure 25; GE Healthcare) and separated using the reverse phase chromatography column (RESOURS RPC 3 mL; GE Healthcare) using the gradient of buffer B [1.0% formic acid, 2.0% acetonitrile] and buffer C [1% formic acid, 70% acetonitrile]. The flow rate was set to 1.0 mL min$^{-1}$ and the column was equilibrated with 75% buffer B and 25% buffer C. Then, the amount of buffer C was increased to 75% over a 30 min period (10-time the column volume). PRs were eluted with 35–50% buffer C. After the elution, buffer C amount was increased to 100% in 6 min and returned to the starting condition over the next 6 min. The peak fractions including PR were collected and three-time diluted with buffer B. The diluted PR solution was injected into the ÄKTA pure 25 again and the targeted PR was purified using the same purification step as described above. The collected fractions containing PR were subjected to desalting (HiTrap Desalting; GE Healthcare) and the eluted solution was equilibrated using 100 mM formic acid and stored at −80°C until use.

## Crystallization

The unfolded PR$^{WT}$ was refolded with the addition of a neutralizing buffer A [100 mM ammonium acetate pH 6.0, 0.005% Tween-20], making the final pH 5.0 to 5.2. The PR$^{WT}$-containing solution was run through Amicon Ultra-15 10K centrifugal filter units (Millipore), giving a solution containing PR (5 ~ 8 mg/ml) in 10 mM ammonium acetate pH 5.0% and 0.005% Tween-20. Occasionally, twice greater concentrations of a test compound were used for crystallization. After centrifugation, the supernatants were collected and subjected to crystallization using the hanging-drop vapor diffusion method. Nextal Tubes ProComplex Suite (QIAGEN) and Wizard Crystallization Screen Series (Emerald BioSystems) were used for the first screening to determine the optimum crystallization condition. PR$^{WT}$-GRL-142 complexes were formed in 0.1 M sodium/potassium phosphate pH 6.2, 20% (weight/volume) polyethylene glycol 1000, 0.2 M sodium chloride. PR$^{WT}$-GRL-121 complexes were formed in 0.1 M 2-morpholinoethanesulfonic acid pH6.5, 25% (weight/volume) polyethylene glycol 4000, 0.2 M potassium iodide. Crystals of PR$^{WT}$ complexed with GRL-121 or GRL-142 was retrieved, immersed into cryoprotective solution containing reservoir solution plus 30% glycerol, and flash-frozen in liquid nitrogen.

## X-ray diffraction data collection and processing details

X-ray diffraction data for PR$^{WT}$-GRL-121 and PR$^{WT}$-GRL-142 were collected at beamline BL41XU located at the SPring-8, Japan. The source wavelength for this data collection was 1.0 Å. The exposure rate per frame was 0.5 s with a frame width of 0.5°. The diffraction images were captured using

DECTRIS PILATUS3 6M detector that was set at distances of 250 mm and 210 mm from the crystals of PR$^{WT}$-GRL-121 and PR$^{WT}$-GRL-142, respectively. Diffraction data were processed and scaled using HKL2000 (*Otwinowski and Minor, 1997*). Details of the X-ray diffraction data collection and processing parameters are given in *Table 7*.

## Structure solutions and refinement

Structure solutions were obtained using molecular replacement (MR) as described previously (*Yedidi et al., 2013*). Briefly, the X-ray crystal structure of PR$^{WT}$ taken from PDB (RRID: SCR_012820) ID: 4HLA was used as a search model to obtain structure solutions through MOLREP, CCP4 (RRID: SCR_007255) (*Vagin and Teplyakov, 1997*; *Winn et al., 2011*). REFMAC5(43) was used to refine the structure solutions through the CCP4 interface. Initial coordinates for GRL-121 and GRL-142 were built in Maestro (Ver. 10.5.014, Schrodinger LLC, New York, NY) and the geometry was optimized using REEL, the ligand restraints viewer and editor in PHENIX (RRID: SCR_014224) (Ver. 1.11.1–2575) (*Adams et al., 2010*). Ligand refinement libraries for GRL-121 and GRL-142 were optimized using the *electronic* ligand builder and optimisation workbench (eLBOW)(*Moriarty et al., 2009*) on the NIH-Biowulf Linux cluster (RRID: SCR_007169). Geometry optimized GRL-121 and GRL-142 were fit into their respective difference electron density maps using ARP/wARP ligands (*Lamzin and Wilson, 1993*; *Zwart et al., 2004*) followed by building the solvent molecules using the ARP/wARP solvent-building module through the CCP4 interface. The final models including the ligands and solvent were refined using the simulated annealing method from phenix.refine (PHENIX, Ver. 1.11.1–2575) on the NIH-Biowulf Linux cluster. Details of the structure refinement statistics are given in *Table 8*.

## Structural analysis

The final refined structures were used for structural analysis. Hydrogen atoms were added using the protein preparation wizard via Maestro (Ver. 10.5.014, Schrodinger LLC, New York, NY). The orientations of added hydrogen atoms were optimized by sampling the orientations of the solvent molecules at pH 7.0 using PROPKA. Hydrogen bonds were calculated using a maximum distance cutoff value of 3.0 Å between the donor and acceptor heavy atoms with minimum angles being 90° (donor) and 60° (acceptor). Hydrophobic contacts were calculated between two carbon atoms (one from the

**Table 7.** X-ray diffraction data processing details for PR$^{WT}$ in complex with GRL-121 or −142.

|  | PR$^{WT}$ + GRL-121 | PR$^{WT}$ + GRL142 |
|---|---|---|
| PDB entry | 5TYR | 5TYS |
| Resolution range (Å) | 50.0–1.7 | 50.0–2.0 |
| Unit cell - a (Å) | 62.56 | 63.13 |
| b (Å) | 62.56 | 63.13 |
| c (Å) | 82.66 | 82.23 |
| α (°) | 90 | 90 |
| β (°) | 90 | 90 |
| γ (°) | 120 | 120 |
| Space group | $P6_1$ | $P6_1$ |
| Solvent content (%) | 54.22 | 54.8 |
| No. of unique reflections | 20,171 (994)* | 12,475 (624) |
| Mean ($I/\sigma(I)$) | 29.05 (3.3) | 26.6 (4.8) |
| $^{\dagger}R_{merge}$ | 0.09 (0.55) | 0.10 (0.46) |
| Data redundancy | 10 (10.1) | 10.2 (9.7) |
| Completeness (%) | 100 (100) | 99.9 (100) |

*Values in parentheses are for the highest resolution shell

$^{\dagger}R_{merge} = \Sigma \ |I - <I> | \ / \ \Sigma \ I$

DOI: https://doi.org/10.7554/eLife.28020.013

**Table 8.** Refinement statistics for structure solutions of PR$^{WT}$ in complex with GRL-121 or −142.

| | PR$^{WT}$ + GRL-121 | PR$^{WT}$ + GRL-142 |
|---|---|---|
| PDB entry | 5TYR | 5TYS |
| Resolution range (Å) | 45.32–1.8 | 32.8–2.01 |
| No. of reflections used | 17,043 | 12,421 |
| *$R_{cryst}$ | 0.191 | 0.1949 |
| $R_{free}$ | 0.232 | 0.2366 |
| No. of protease dimers per †AU | 1 | 1 |
| No. of protein atoms per AU | 1516 | 1516 |
| No. of ligand molecules per AU | 2 | 2 |
| No. of ligand atoms per AU | 92 | 96 |
| No. of water molecules | 148 | 90 |
| Mean temperature factors: | | |
| Protein (Å$^2$) | 24.21 | 28.36 |
| Main chains (Å$^2$) | 22.02 | 25.84 |
| Side chains (Å$^2$) | 26.6 | 31.11 |
| Ligand (Å$^2$) | 16.55 | 20.05 |
| Waters (Å$^2$) | 34.24 | 34.9 |
| RMSD bond lengths (Å) | 0.007 | 0.008 |
| RMSD bond angles (Å) | 1.015 | 1.022 |
| Ramachandran plot: | | |
| Most favored (%) | 99.48 | 97.94 |
| Additional allowed (%) | 0.52 | 2.06 |
| Generously allowed (%) | 0 | 0 |
| Disallowed (%) | 0 | 0 |

*$R_{cryst} = \Sigma\ ||F_{obs}| - |F_{calc}||\ /\ \Sigma|F_{obs}|$
†AU - Asymmetric unit
DOI: https://doi.org/10.7554/eLife.28020.014

PI and one from PR$^{WT}$) with a maximum 4 Å distance cutoff. The van der Waals contact, C is defined as D12 / (R1 +R2) where D12 is the distance between atomic centers 1 and 2; R1, R2 are the van der Waals radii of atomic centers 1 and 2. C has a value between 0.89 and 1.30 for favorable contacts. To calculate the average van der Waals interaction energy, a 1.2 ns molecular dynamics simulation of GRL-142 or DRV in complex with HIV-1 protease was carried out in explicit water with OPLS3 force field, and the trajectories were analyzed. The simulation program DESMOND (RRID: SCR_014575) (D.E. Shaw Research, New York, NY 2017) was used.

## Thermal stability analysis using differential scanning fluorimetry (DSF)

Each PR tested (pH 2.8) was refolded by adding 100 mM ammonium acetate (pH 6.0), which resulted in the final pH of 5.0 to 5.2. Then the Tween-20 was added to the refolded PR solution (0.01% Tween-20) and the solution was concentrated using Amicon Ultra-15 10K centrifugal filter units (Millipore). In conducting DSF analysis, the final concentration of each PR used was 10 μM (dimer) and the final compound concentration used was 50 μM. Recombinant human renin was purchased from ProSpec-Tany Technogenic (Ness-Ziona, Israel). Lysozyme was purchased from Affymetrix (Santa Clara, CA). The final concentration of human renin and lysozyme were 5 μM, respectively. In the experiment, 20 mM compound solutions in dimethyl sulfoxide (DMSO) were used as stock solutions. The final buffer used contained 100 mM ammonium acetate pH 5.0, 0.01% Tween-20, SYPRO Orange (5X)(Life Technologies) and 2.5% DMSO with 50 mM sodium chloride for TFR-PR$^{D25N-7AA-His6}$, Human Renin, and Lysozyme or without 50 mM sodium chloride for PR$^{D25N}$. Thirty microliter

solution was successively heated from 15°C to 95°C, and the changes of fluorescence intensity were documented using the real-time PCR system (Applied Biosystems).

## Electrospray Ionization mass spectrometry

DRV and GRL-142 were dissolved in methanol (2.5 mM). Unfolded PR[D25N] in 100 mM formic acid (pH 2.8) was refolded in the presence of DRV or GRL-142 (50 µM) by the addition of 100 mM ammonium acetate (pH 6.0) containing the same concentration of the compound, making the final pH to 5.0 to 5.2 and the preparation was subjected to centrifugation and the supernatants were collected. The MS spectra of PR[D25N] with or without compound were obtained using a Bio-Tof-Q electrospray ionization (ESI) quadrupole time-of-flight mass spectrometer (Bruker Daltonics). Each sample solution was introduced to the Bio-Tof-Q through an infusion pump at a flow rate of 8 µL/min. Measurement conditions were as follows: dry $N_2$ gas temperature, 100°C; ESI, positive mode; capillary voltage, 3500 V; voltages of capillary exit and skimmer, 80 V and 45 V, respectively. The deconvoluted spectra of the protein ions were obtained using the data analysis software version 3.1 (Bruker Daltonics).

## Preparation of PBMCs extracts for cellular uptake assays

Cell extracts were prepared as previously described (*Yedidi et al., 2013*). Briefly, PBMCs ($5 \times 10^6$) were incubated with each compound (0.1, 1 and 10 µM final concentrations) at 37°C for 60 min. Cells were then harvested and washed with PBS three times. The final cell pellets were resuspended in 60% methanol, and the suspensions were incubated at 95°C for 5 min with shaking. The prepared suspensions were cooled to room temperature and centrifuged at 13,000 rpm for 10 min to separate the solvent extract from cell debris. Supernatants (solvent extracts) were transferred into new tubes, and the solvent was evaporated overnight. DMSO (50 µL) was added to the dried preparations and incubated at 37°C for 1 hr with shaking. Samples were then analyzed for quantification of the amount of each compound using TOF LC/MS.

## TOF LC/MS analysis of protease inhibitors

Each preparation (20 µl) was subjected to Agilent TOF LC/MS. Each compound was separated through VyDac C18 5-µm-particle-size column (1.0 mm by 50 mm) using gradient of solvent A (water–0.01% formic acid) and solvent B (acetonitrile–0.1% formic acid). The flow rate was set to 0.5 ml min$^{-1}$, and the column was equilibrated with 95% solvent A and 5% solvent B. Following each injection, solvent B was increased to 55% over a 20 min period (2.5% increase per min). At 21 min solvent B was increased to 95% in 1 min and then returned to starting conditions over the next 1 min. Each compound was detected by TOF-MS using an Agilent 6230 mass spectrometer in selective ion monitoring mode. The sodium adducts of each compound provided the most prominent peak and therefore were used for detection purposes although the parent ions provided the relative results. The amount of compound obtained in the extracts was determined by comparison to standards of each purified compound dissolved in DMSO. Each compound was confirmed with both elution time over the column and molecular weight by mass spectrometry.

## Determination of GRL-142 concentrations in CNS in rats

Studies using rats were carried out according to the Guidelines for Animal Studies defined by the Committee for Animal Care and Use of Nonclinical Research Center, LSI Medience Corporation. The experimental protocol was approved by the Committee of Nonclinical Research Center, LSI Medience Corporation (approval No. 2016–0039). In the examination of CNS-penetration capability of GRL-142, male Sprague-Dawley rats (8 weeks) were obtained from Charles River Laboratories Japan, Inc. For oral injection, test compounds were dissolved in 50% propylene glycol (Wako Pure Chemical Industries, Ltd., Osaka, Japan) and 17% (v/v) DMSO. Prepared solutions containing GRL-142 (5 mg/kg) plus RTV (8.33 mg/kg) or DRV (5 mg/kg) plus RTV (8.33 mg/kg) were perorally administered to four rats. Plasma, cerebrospinal fluid (CSF), and brain were collected from each rat at indicated time points following the drug administration. Whole brains excised out were carefully washed with PBS and then wiped with filter paper to minimize blood contamination until blood was not visually seen and stored at −80°C until analyzed. Plasma and CSF samples were also stored at −80°C until use.

GRL-142 or DRV was extracted from the plasma, CSF, and brain samples by a solid-phase extraction method using Oasis MCX µElution plates (Waters Corporation, Milford, MA, USA). GRL-142

calibration standards ranging from 0.1 to 100 ng/mL were prepared using standard solutions with rat blank plasma, CSF, and brain. Linearity of the calibration curves were defined by the following formulae: Accuracy (%)=Mean measured value/Nominal value ×100; Precision (%)=Standard deviation/Mean measured value ×100. Acceptable criteria of the accuracy were set up to be within 100 ± 15% and the precision 15% or below. Saquinavir mesylate (Toronto Research Chemicals Inc., Toronto Canada) was used as an internal standard (IS).

Chromatographic separation of GRL-142 from the plasma, CSF, and brain samples was performed using Acquity Ultra Performance LC (Waters Corporation, Milford, MA, USA) and an Acquity UPLC BEH C18 column (2.1 × 5.0 mm, 1.7 μm) maintained at 40°C. Mobile phases A and B consisted of 10 mM ammonium formate aqueous solution and acetonitrile, respectively. Separation was performed using gradient elution at a flow rate of 0.7 mL/min. Quantitation of each compound was conducted by selected reaction monitoring on a API5000 mass spectrometer (AB SCIEX, Framingham, MA, USA) with electrospray ionization in the positive mode. The optimized electrospray ionization parameters were as follows: ion source temperature, 650°C; curtain gas, setting 10; nebulizing gas (GS1), setting 40; heater gas (GS2), setting 50; ion spray voltage, 5500 V. The selected reaction monitoring transitions were m/z 707 to 253 for GRL-142, m/z 548 to 392 for DRV, and m/z 672 to 570 for IS. The dwell time was 150 ms for each transition channel. All data were acquired and analyzed using the Analyst version 1.5 software (AB SCIEX, Framingham, MA, USA).

## Acknowledgements

The present work was supported in part by the Intramural Research Program of the Center for Cancer Research, National Cancer Institute, National Institutes of Health (HM and RY); a grant from the National Institutes of Health (GM53386; AKG); a grant for Development of Novel Drugs for Treating HIV-1 Infection and AIDS from Japan Agency for Medical Research and Development (HM); grants from Japan Society for the Promotion of Sciences; and a grant from National Center for Global Health and Medicine Research Institute. The authors also thank the synchrotron beam line staff at SPring-8 for their support in X-ray diffraction data collection and the support by the Platform Project for Supporting in Drug Discovery and Life Science Research from the Ministry of Education, Culture, Sports, Science and Technology (MEXT). This study utilized the high-performance computational capabilities of the Biowulf Linux cluster at the National Institutes of Health, Bethesda, MD (http://hpc.nih.gov).

## Additional information

### Funding

| Funder | Author |
| --- | --- |
| National Institutes of Health | Robert Yarchoan<br>Arun K Ghosh<br>Hiroaki Mitsuya |
| Japan Agency for Medical Research and Development | Hiroaki Mitsuya |
| Japan Society for the Promotion of Science | Hiroaki Mitsuya |
| National Center for Global Health and Medicine Research Institute | Hiroaki Mitsuya |

The funders had no role in study design, data collection and interpretation, or the decision to submit the work for publication.

### Author contributions

Manabu Aoki, Conceptualization, Data curation, Formal analysis, Validation, Investigation, Writing—original draft, Writing—review and editing; Hironori Hayashi, Nobuyo Higashi-Kuwata, Hiromi Aoki-Ogata, Data curation, Formal analysis, Validation, Investigation, Writing—original draft; Kalapala

Venkateswara Rao, David A Davis, Prasanth R Nyalapatla, Heather L Osswald, Shogo Misumi, Data curation, Formal analysis, Validation, Investigation; Debananda Das, Data curation, Formal analysis, Validation, Investigation, Visualization, Writing—original draft, Writing—review and editing; Haydar Bulut, Data curation, Formal analysis, Validation, Investigation, Visualization, Writing—original draft; Yuki Takamatsu, Shin-ichiro Hattori, Kazuya Hasegawa, Nobutoki Takamune, Data curation, Formal analysis, Investigation; Ravikiran S Yedidi, Data curation, Formal analysis, Investigation, Writing—original draft; Noriko Nishida, Data curation, Formal analysis, Validation, Investigation, Methodology; Hirofumi Jono, Hideyuki Saito, Data curation, Formal analysis, Methodology, Writing—review and editing; Robert Yarchoan, Supervision, Funding acquisition, Project administration, Writing—review and editing; Arun K Ghosh, Conceptualization, Resources, Funding acquisition, Validation, Writing—review and editing; Hiroaki Mitsuya, Conceptualization, Resources, Supervision, Funding acquisition, Validation, Methodology, Project administration, Writing—review and editing

**Author ORCIDs**
Debananda Das [ID] http://orcid.org/0000-0001-9943-5888
Yuki Takamatsu [ID] http://orcid.org/0000-0001-7967-1576
Hiroaki Mitsuya [ID] http://orcid.org/0000-0001-9274-3853

**Ethics**
Animal experimentation: Studies using rats were carried out according to the Guidelines for Animal Studies defined by the Committee for Animal Care and Use of Nonclinical Research Center, LSI Medience Corporation. The experimental protocol was approved by the Committee of Nonclinical Research Center, LSI Medience Corporation (approval No. 2016-0039).

**Decision letter and Author response**
Decision letter https://doi.org/10.7554/eLife.28020.019
Author response https://doi.org/10.7554/eLife.28020.020

# Additional files

**Supplementary files**
• Supplementary file 1. Antiviral activity of five PIs and four NRTIs against laboratory-selected PI-resistant HIV-1 variants.
DOI: https://doi.org/10.7554/eLife.28020.015

• Supplementary file 2. Representative dose-response profiles of DRV, GRL-121, and GRL-142 against $cHIV_{NL4-3}^{WT}$, $cHIV_{NL4-3}^{V32I}$, $cHIV_{NL4-3}^{G48V}$, $cHIV_{NL4-3}^{I50V}$, and $cHIV_{NL4-3}^{V82T}$ are shown.
DOI: https://doi.org/10.7554/eLife.28020.016

• Supplementary file 3. List of 37 HIVs used in antiviral assay and the amino acid sequences of the protease-encoding region of HIVs used in this study.
DOI: https://doi.org/10.7554/eLife.28020.017

• Transparent reporting form
DOI: https://doi.org/10.7554/eLife.28020.018

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
