## [Decision Letter]

Thank you for submitting your article "A Novel CNS-Penetrating Protease Inhibitor Overcomes HIV-1 Resistance with Unprecedented aM to pM Potency" for consideration by *eLife*. Your article has been reviewed by two peer reviewers, and the evaluation has been overseen by Wes Sundquist as Reviewing Editor and by Arup Chakraborty as the Senior Editor. The following individual involved in review of your submission has agreed to reveal his identity: Tomas Cihlar (Reviewer #3).

The reviewers have discussed the reviews with one another and the Reviewing Editor has drafted this decision to help you prepare a revised submission.

Summary:

The submitted manuscript is focused on novel modifications of HIV protease inhibitors (PIs) and represents a very significant advancement in the design of highly potent compounds, particularly in case of GRL-142, which is the most potent PI reported to date. The described compounds are derivatives of darunavir (DRV) with unique modifications in P1, P2, and/or P2' positions that confer exquisite antiviral potency in vitro against HIV. In addition, high lipophilicity of these compounds may allow for improved penetration across blood-brain barrier. The inhibitor also retains high level activity against highly drug resistant variants of HIV-1 protease that exhibit high levels of resistance to DRV. It should also be taken into account, however, that multiple aspects of pharmacological profiling of these novel compounds remain to be assessed, and their full potential for further development towards products useful for the treatment of HIV infection therefore remains to be determined.

The following issues need to be addressed prior to publication:

1) The authors have recently published 3 of the compounds (GRL-036 (compound 27), GRL-139 (compound 4) and GRL-121 (compound 5)) presented in this manuscript with the "Crown-Like Oxotricyclic Core" in J Med Chem 2017 60, 4267-4278. Antiviral data against many of the same variants was presented as were crystal structures including 5TYR. This paper is not referenced at all – and must have been at least "in press" when this manuscript was submitted.

2) Although the title claims that these inhibitors are aM (10-18) in potency, it is unclear whether measurements of this level of potency are actually possible. Curves showing how many points are measured in the inflection to determine such high potency seem essential. The errors are of similar magnitude as the values and many single site mutations appear to have orders of magnitude higher potency than the wildtype (Table 4).

3) Abstract: The claim that current HIV therapies are not well tolerated and cause a lot of resistance needs to be changed because it is not accurate.

4) Subsection “Identification of GRL-142, an unprecedentedly potent HIV-1 protease inhibitor”, second paragraph and Table 1: The high potency of GRL-142 against DRV-resistant HIV-1 variants is very encouraging. However, the authors should acknowledge that these viruses are still substantially less sensitive to GRL-142 (up to 63-fold) than the wild-type virus, indicating considerable resistance shift.

5) Inhibition of dimerization (Figure 2) is unclear as to whether the differences are statistically significant. Also in terms of the anti-viral activity – how could the subtle difference in potential dimerization inhibition – experiments performed in the high μM concentration range – have a tangible impact on the antiviral activity? What was the control experiment to eliminate a possibility of GRL-142 quenching the dimerization-dependent FRET signal rather than truly inhibiting the dimerization? In addition, what is the potency of GRL-142 on inhibiting the enzymatic activity of PR relative to the inhibition of dimerization? Finally, please also explicitly list the variants used in the antiviral assays.

6) Analysis of contact (Figure 3) as "numbers of contacts" is a very qualitative measure of packing, better would be to calculate the van der Waals energies. The figures in J Med Chem paper were much better representations of the packing and what residues are potentially contributing to affinity. Along similar lines it would be very useful to know which of the residues that evolve in the viral selection are packing against the inhibitor or are remote from the active site?

7) Subsection “Identification of GRL-142, an unprecedentedly potent HIV-1 protease inhibitor”, last paragraph and Table 4: What is the explanation for the extreme hypersensitivity of some of the single residue mutant viruses to GRL-142 (e.g., V32I, G48V, and I50V)? This hypersensitivity reaches up to 1,000,000-fold relative to wild-type HIV-1, and such single mutation hypersensitivity seems unprecedented, especially in the case of I50V, which is one of the primary resistance mutations for DRV. This observation raises questions about the activities of the activity measurements.

8) Subsection “CNS penetration of GRL-142 in rats”: High resistance barrier for GRL-142. While the authors described the genotypic analysis of viruses propagated in the presence of increasing concentrations of GRL-142 (i.e., HIV-DRV-R-P30-142-W36 and rCL^-^HIV-T48-142-W36), no data are presented on the phenotypic sensitivity of these viruses that can propagate in the presence of increasing GRL-142 concentrations. While the authors argue these concentrations were not numerically very high, they still exceeded the IC_50_ values by >100fold in some cases. Thus these selected variants should be analyzed for GRL-142 susceptibility shift.

9) Subsection “CNS penetration of GRL-142 in rats”, CNS penetration: "The plasma concentrations of DRV were much greater than the IC_50_ value (~3.2 μM)" This is probably a typo in concentration unit as the IC_50_ of DRV is in low nM, not μM range? How do the authors explain undetectable levels of GRL-142 in CSF, yet accumulation in brain? This is rather unexpected since there should be equilibrium between CSF and brain tissue similar to the equilibrium between plasma and PBMCs or lymphoid tissue. Is it possible that the highly lipophilic nature of GRL-142 results in accumulation of the molecule in brain lipid fraction and not the free unbound fraction, making the compound not available to inhibit HIV replication in brain?

---

## [Author Response]

The following issues need to be addressed prior to publication:1) The authors have recently published 3 of the compounds (GRL-036 (compound 27), GRL-139 (compound 4) and GRL-121 (compound 5)) presented in this manuscript with the "Crown-Like Oxotricyclic Core" in J Med Chem 2017 60, 4267-4278. Antiviral data against many of the same variants was presented as were crystal structures including 5TYR. This paper is not referenced at all – and must have been at least "in press" when this manuscript was submitted.

The report on three compounds (GRL-036, GRL-139, and GRL-121) heavily focused on their chemistry and detailed synthetic methods (J Med Chem 60: 4267-4278, 2017) and the present manuscript focuses on GRL-142’s biological/virological features. The finalization of the present paper for us to submit for publication took much longer than we anticipated. To our regret, we failed to cite the J Med Chem paper by Ghosh et al. when we submitted the present paper on GRL-142 for possible publication in eLife. We have now cited the J Med Chem paper by Ghosh et al. in the revised version of the manuscript

2) Although the title claims that these inhibitors are aM (10-18) in potency, it is unclear whether measurements of this level of potency are actually possible. Curves showing how many points are measured in the inflection to determine such high potency seem essential. The errors are of similar magnitude as the values and many single site mutations appear to have orders of magnitude higher potency than the wildtype (Table 4).

As pointed out by the reviewer, there are some technical challenges involved in diluting the working solution (2 µM) of GRL-142 down to femtomolar – attomolar concentrations. A large number of serial dilution causes variability of data due to (i) the adsorption of the compound to the walls of plastic pipets and the plastic surface of microtiter culture plate wells, where serial dilution was directly performed, resulting in the relative loss of the compound adsorbed to the wells and (ii) the carry-over of the compound adsorbed to the walls of pipets to the next microtiter culture plate wells, which could result in the increased delivery of the compound to the next microtiter culture wells. We do not encounter such inherent problems when we handle less potent compounds such as darunavir with an IC_50_ value of ~3 nM. Thus, in the initial version of the manuscript, we provided the method of dilution in detail. Nevertheless, we confirmed the reproducibility of the data by repeating the experiments at least three to six times upon different occasions in triplicate with one HIV-1 species.

In Supplementary file 2 of the revised version of the manuscript, as per reviewer’s suggestion, we have added the virus titration curves of DRV, GRL-121, and GRL-142 against cHIV_NL4-3_^WT^, cHIV_NL4-3_^V32I^, cHIV_NL4-3_^G48V^, cHIV_NL4-3_^I50V^, and cHIV_NL4-3_^V82T^. GRL-142’s HIV-1 titration curves against cHIV_NL4-3_^V32I^, cHIV_NL4-3_^G48V^, and cHIV_NL4-3_^I50V^ show that GRL-142 exerts significantly greater antiviral activity against the HIV-1 variants than against cHIV_NL4-3_^WT^.

3) Abstract: The claim that current HIV therapies are not well tolerated and cause a lot of resistance needs to be changed because it is not accurate.

As suggested, we have changed the sentence in question as follows: “However, certain individuals who initially achieve viral suppression to undetectable levels may eventually suffer treatment failure mainly due to adverse effects and the emergence of drug-resistant HIV-1 variants.”

4) Subsection “Identification of GRL-142, an unprecedentedly potent HIV-1 protease inhibitor”, second paragraph and Table 1: The high potency of GRL-142 against DRV-resistant HIV-1 variants is very encouraging. However, the authors should acknowledge that these viruses are still substantially less sensitive to GRL-142 (up to 63-fold) than the wild-type virus, indicating considerable resistance shift.

As per the reviewer’s suggestion, we now mention that the most DRV-resistant HIV-1 variant, HIV_DRV_^R^_P51_, is less sensitive to GRL-142 by 63-fold than cHIV_NL4-3_^WT^, indicating considerable resistance shift in the revised version of the manuscript as follows:

“[…] while it should be noted that the most DRV-resistant HIV-1 variant, HIV_DRV_^R^_P51_, is less sensitive to GRL-142 by 63-fold than HIV_NL4-3_, indicating considerable resistance shift.”

5) Inhibition of dimerization (Figure 2) is unclear as to whether the differences are statistically significant. Also in terms of the anti-viral activity – how could the subtle difference in potential dimerization inhibition – experiments performed in the high μM concentration range – have a tangible impact on the antiviral activity? What was the control experiment to eliminate a possibility of GRL-142 quenching the dimerization-dependent FRET signal rather than truly inhibiting the dimerization? In addition, what is the potency of GRL-142 on inhibiting the enzymatic activity of PR relative to the inhibition of dimerization? Finally, please also explicitly list the variants used in the antiviral assays.

i) Statistical difference in the intermolecular FRET-based HIV-1 expression assay.

We determined and described the p-values of CFP^A/B^ ratios in the absence and presence of each test compound using the Wilcoxon rank-sum test (JMP software, SAS, Cary, NC) in the legends to Figure 2 of the initial version of the manuscript. As can be seen, the CFP^A/B^ ratios in the presence of 0.1 nM GRL-142 were less than 1.0 (p=0.0585) than in the absence of the compound. The p values of the difference between the ratios in the presence of 1, 10, 100, and 1,000 nM compared to those in the absence of GRL-142 were 0.0145, 0.0042, 0.0056, and 0.0019, respectively. We have described these statistics in the Results section of the revised version of the manuscript.

ii) Impact of the PR dimerization inhibition on GRL-142’s potent anti-HIV-1 activity.

GRL-142 inhibited HIV-1 protease (PR) dimerization at 0.1 nM and greater concentrations (Please note that the unit is nM, *not* µM: Figure 2), which is close to GRL-142’s IC_50_ value (0.017 nM) determined in the cell-based HIV-1 replication inhibition assays. However, we cannot directly compare those values since the numbers of the target PR molecules in COS7 are thought to be different from those in the target MT-4 cells exposed to HIV-1. In the FRET-HIV-1 expression assay, the cells are transfected with plasmids encoding an entire length of HIV-1 and the virions are transiently expressed/produced over just ~3 days of the culture following the transfection. In contrast, in the cell-based assays, the target MT-4 cells exposed to HIV-1 continue to express/produce virions over the entire 7-day period of the culture. Nevertheless, the potent PR dimerization inhibition of GRL-142 is thought to explain at least in part the mechanism of the potent anti-HIV-1 activity of GRL-142. We have described this issue in the revised version of the manuscript.

iii) Control in the FRET-HIV-1 expression assay.

We previously demonstrated that two PR inhibitors, darunavir (DRV) and tipranavir (TPV), block PR dimerization in the FRET-HIV-1 expression assay (Koh and Mitsuya et al. JBC 2007 and Aoki and Mitsuya et al. JV 2012). Although we have no particular assay system to ask whether PR inhibitors such as DRV and GRL-142 quench the dimerization-dependent FRET signals, we have previously shown that when an HIV_DRV_^R^_P51_-derived HIV-1 clone containing 14 amino acid substitutions (L10I, I15V, K20R, L24I, V32I, L33F, M36I, M46L, I54M, L63P, K70Q, V82I, I84V, and L89M) was employed in the FRET-HIV-1 expression assay instead of HIV_NL4-3_, even a relatively high concentration of DRV (100 nM) allowed the occurrence of FRET (Koh JV 2011), strongly suggesting that 100 nM DRV does not quench the FRET signals. Also, other “conventional” PIs (that have no detectable dimerization inhibition) such as saquinavir, lopinavir, nelfinavir, amprenavir, and atazanavir at high concentrations (1 and 10 µM) allowed the occurrence of FRET (Koh and Mitsuya et al. JBC 2007), suggesting there was no quenching. However, the FRET-HIV-1 expression system offers only qualitative (*not* quantitative) analysis. Thus, we also asked if DRV directly blocks PR dimerization using electrospray ionization mass spectrometry (ESI-MS) and demonstrated that DRV binds to PR monomers in a one-to-one molar ratio, inhibiting the first step of PR dimerization, whereas conventional protease inhibitors (such as saquinavir) inhibits enzymatic activity but not dimerization and fails to bind to monomers (Hayashi and Mitsuya et al. PNAS 2014). In the present manuscript, we demonstrate that GRL-142 more tightly binds to PR monomers than DRV, corroborating that GRL-142 truly blocks PR dimerization (Figure 2).

iv) The potency of GRL-142 on inhibiting the enzymatic activity of PR relative to the inhibition of dimerization?

As we stated in our response to reviewer’s comment 5 (ii) above, the intensity of PR dimerization inhibition is determined using COS7 cells that are transfected with plasmids encoding an entire length of HIV-1 and the virions are transiently expressed/produced in the cytoplasm over ~3 days of the culture following the transfection in the presence or absence of PIs. In contrast, enzymatic inhibition is determined using a solution of purified protease. Thus, it is difficult to stoichiometrically compare the data generated in two different assays. However, it should be certainly adequate to compare the potency levels of dimerization inhibition by DRV and GRL-142. At this time, HIV-1 PR dimerization inhibitors such as DRV, TPV, and GRL-142 also inhibit HIV-1 PR enzymatic activity. If we get HIV-1 PR dimerization inhibitors that do not have PR enzyme inhibition activity, we should be able to more precisely determine the impact of PR dimerization inhibition activity on anti-HIV-1 activity relative to the impact of PR enzyme inhibition activity.

v) Explicitly list the variants used in the antiviral assays.

As per the reviewer’s suggestion, we have provided a list of viruses used in antiviral assay as Supplementary file 3 in the revised manuscript.

6) Analysis of contact (Figure 3) as "numbers of contacts" is a very qualitative measure of packing, better would be to calculate the van der Waals energies. The figures in J Med Chem paper were much better representations of the packing and what residues are potentially contributing to affinity. Along similar lines it would be very useful to know which of the residues that evolve in the viral selection are packing against the inhibitor or are remote from the active site?

According to the suggestion, the figures have been modified. Figure 3 has been deleted, and Figure 3 has been revised to show contacts that make more chemical sense by only showing significant interactions and not showing the non-polar hydrogens. Figure 3 has been updated to show the interactions of the Connolly surfaces. Moreover, we have calculated the van der Waals energies of interactions between GRL-142 and several active site residues and compared those with the corresponding interactions of DRV (Table 5 of the revised version of the manuscript). The van der Waals energies have been discussed in the manuscript.

We have also added additional figures (Figure 3), along the lines of our figure in the J. Med. Chem. that shows the interactions of GRL-142 and DRV with the active site residues.

7) Subsection “Identification of GRL-142, an unprecedentedly potent HIV-1 protease inhibitor”, last paragraph and Table 4: What is the explanation for the extreme hypersensitivity of some of the single residue mutant viruses to GRL-142 (e.g., V32I, G48V, and I50V)? This hypersensitivity reaches up to 1,000,000-fold relative to wild-type HIV-1, and such single mutation hypersensitivity seems unprecedented, especially in the case of I50V, which is one of the primary resistance mutations for DRV. This observation raises questions about the activities of the activity measurements.

As we stated above in our response to reviewer’s comment 2, we have added the HIV-1 replication inhibition titration curves of DRV, GRL-121, and GRL-142 against cHIV_NL4-3_^WT^, cHIV_NL4-3_^V32I^, cHIV_NL4-3_^G48V^, cHIV_NL4-3_^I50V^, and cHIV_NL4-3_^V82T^ in Supplementary file 2 of the revised version of the manuscript, where one can see that GRL-142’s HIV-1 titration curves against cHIV_NL4-3_^V32I^, cHIV_NL4-3_^G48V^, and cHIV_NL4-3_^I50V^ show GRL-142’s significantly greater antiviral activity against those HIV-1 variants than against cHIV_NL4-3_^WT^. In terms of the reproducibility of the data, as we noted above, the assays were repeatedly conducted at least three to six times upon different occasions in triplicate with one HIV-1 species.

The mechanism of such an extreme hypersensitivity of those variants with a single amino acid substitution remains to be elucidated. In regard to the hypersensitivity of cHIV_NL4-3_^V32I^ against GRL-142, the Connolly surfaces of Ile32 and Ile32' have better interactions with GRL-142 than the corresponding valines (Figure 3). The van der Waals (vdW) interaction energy between GRL-142 and Ile32 is lower by 1.1 kcal/mol compared to the corresponding interaction with Val32. This feature should explain at least in part the potent antiviral activity of GRL-142 against cHIV_NL4-3_^V32I^. Of note, substitution of an amino acid in PR may alter interactions not only between the substituted amino acid and a compound but also between amino acids around the substituted amino acid and the compound (Liu, J Mol Biol, 2008; Mittal, J Virol, 2013) and further quantitative analyses remain to be conducted for such binding energy alterations, which is the topic for future research.

8) Subsection “CNS penetration of GRL-142 in rats”: High resistance barrier for GRL-142. While the authors described the genotypic analysis of viruses propagated in the presence of increasing concentrations of GRL-142 (i.e., HIV-DRV-R-P30-142-W36 and rCL^-^HIV-T48-142-W36), no data are presented on the phenotypic sensitivity of these viruses that can propagate in the presence of increasing GRL-142 concentrations. While the authors argue these concentrations were not numerically very high, they still exceeded the IC_50_ values by >100fold in some cases. Thus these selected variants should be analyzed for GRL-142 susceptibility shift.

In response to reviewer’s suggestion, we newly conducted assays for the phenotypic features of cHIV_NL4-3_^WT^ and r_CL_HIV_T48_ selected with GRL-142 over 36 weeks, designated as HIV_NL4-3_^WT142-WK36^ and r_CL_HIV_T48_^142-WK36^ respectively, and generated new Table 6, which has been added to the revised version of the manuscript. HIV_NL4-3_^WT142-WK36^ had acquired resistance to GRL-142 by 15-fold compared to the starting population (cHIV_NL4-3_^WT^), while r_CL_HIV_T48_^142-WK36^ had acquired far greater resistance to GRL-142 by 769-fold by the end of the 36-week selection. We have added these data with interpretation description in the revised version of the manuscript.

9) Subsection “CNS penetration of GRL-142 in rats”, CNS penetration: "The plasma concentrations of DRV were much greater than the IC_50_ value (~3.2 μM)" This is probably a typo in concentration unit as the IC_50_ of DRV is in low nM, not μM range? How do the authors explain undetectable levels of GRL-142 in CSF, yet accumulation in brain? This is rather unexpected since there should be equilibrium between CSF and brain tissue similar to the equilibrium between plasma and PBMCs or lymphoid tissue. Is it possible that the highly lipophilic nature of GRL-142 results in accumulation of the molecule in brain lipid fraction and not the free unbound fraction, making the compound not available to inhibit HIV replication in brain?

We regret that the unit µM was a typographic error and corrected to nM in the revised manuscript.

Regarding the undetectable levels of GRL-142 in CSF, two rats were sacrificed each at 60 and 360 minutes after GRL-142 oral administration and GRL-142 concentrations in plasma, CSF, and brain (after irrigation) were determined. Of note, based on our PK study, we chose 60 and 360 minute time points as the early rise phase and plasma peak phase, respectively. It is possible that at an early time point in 60 minutes, GRL-142 might not have reached the equilibrium between CSF and brain tissues. Indeed, in 360 minutes following oral administration relatively good GR-142 concentrations are seen in CSF. Yet the concentrations in brain were generally higher than in CSF. In this regard, as the reviewer mentioned, it is possible that because of the lipophilic feature of GRL-142, the compound might have accumulated in the lipid fraction of the brain. However, when we quantified the GRL-142 amounts in body’s fat tissues of the same rats sacrificed in 360 minutes after oral administration, no significant GRL-142 amount was present. Thus, the GRL-142 distribution features in the CNS remain to be studied further.